# Ecological traits interact with landscape context to determine bees' pesticide risk

Jessica L. Knapp [1,4] ✉, Charlie C. Nicholson[1], Ove Jonsson[2], Joachim R. de Miranda [3] & Maj Rundlöf [1] ✉

Widespread contamination of ecosystems with pesticides threatens non-target organisms. However, the extent to which life-history traits affect pesticide exposure and resulting risk in different landscape contexts remains poorly understood. We address this for bees across an agricultural land-use gradient based on pesticide assays of pollen and nectar collected by *Apis mellifera*, *Bombus terrestris* and *Osmia bicornis*, representing extensive, intermediate and limited foraging traits. We found that extensive foragers (*A. mellifera*) experienced the highest pesticide risk—additive toxicity-weighted concentrations. However, only intermediate (*B. terrestris*) and limited foragers (*O. bicornis*) responded to landscape context—experiencing lower pesticide risk with less agricultural land. Pesticide risk correlated among bee species and between food sources and was greatest in *A. mellifera*-collected pollen—useful information for future postapproval pesticide monitoring. We provide foraging trait- and landscape-dependent information on the occurrence, concentration and identity of pesticides that bees encounter to estimate pesticide risk, which is necessary for more realistic risk assessment and essential information for tracking policy goals to reduce pesticide risk.

Agricultural intensification includes concomitant reductions in semi-natural areas and increased reliance on pesticides[1,2], threatening beneficial insects such as bees that sustain ecosystem function and services[3,4]. Pesticides have received particular attention due to their widespread use yet sometimes detrimental effects on bee individuals[5], colonies[6,7], populations[8,9] and pollination services[10,11]. As pesticide risk (summed toxicity-weighted concentrations) depends on exposure (the degree to which an organism encounters pesticides at a given time and place) it is vital to determine how bee activity patterns intersect with the occurrence, concentration and identity of pesticides[12].

Pesticide-treated cropland, especially of intensively managed fruit and vegetable crops, can increase the amount and diversity of pesticides in the landscape[13–16]. However, pesticides do not just affect target crops and their pests; they can drift and leach into the surrounding air, soil and water to contaminate non-crop plants[17–21]. Thus, seminatural habitats that could provide refuge from pesticides are more likely to be potential sources of exposure in intensively managed agricultural landscapes[22]. As central place foragers, the reproduction of bees depends on the density and value of food resources within their foraging range[23–26] and the proportion of the foraging range of a bee affected by pesticide use should correlate to their pesticide exposure[15,27,28].

On the basis of the unique and correlated traits of bees, including sociality, communication, colony size, foraging capacity and diet breadth, we describe three sets of foraging traits: 'extensive', 'intermediate' and 'limited' (Fig. 1a). These traits will alter the pesticide exposure of bees in landscapes (Fig. 1b; line intercepts)[29]. For example, extensive foragers may be most exposed as they form large, highly eusocial colonies that communicate profitable, albeit potentially treated, mass-flowering crop resources which they can store for extended periods[30]. On the other hand, limited foragers do not accumulate extensive

[1]Department of Biology, Lund University, Lund, Sweden. [2]Department of Aquatic Sciences and Assessment, SLU Centre for Pesticides in the Environment, Swedish University of Agricultural Sciences, Uppsala, Sweden. [3]Department of Ecology, Swedish University of Agricultural Sciences, Uppsala, Sweden. [4]Present address: Department of Botany, Trinity College Dublin, Dublin, Ireland. ✉e-mail: knappj@tcd.ie; maj.rundlof@biol.lu.se

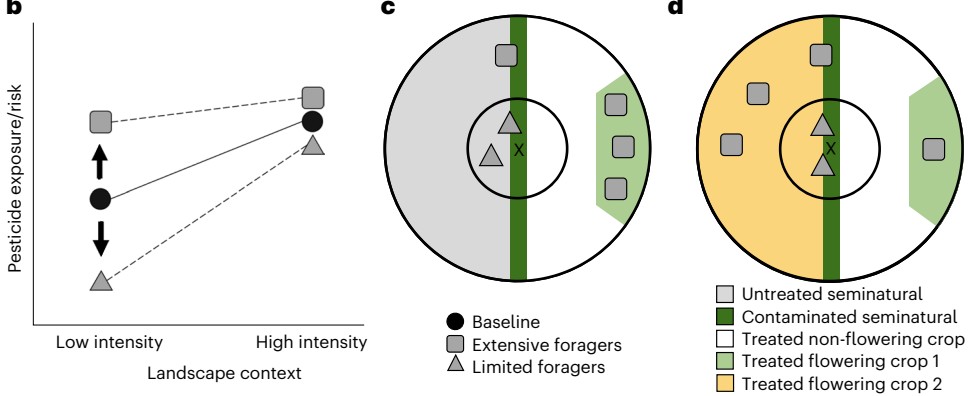

| a | | Limited | Intermediate | Extensive |
|---|---|---|---|---|
| Sociality | | Non-social | Weakly social | Highly social |
| Colony size | | NA | 100s | 10,000s |
| Communication | | None | Primitive | Advanced |
| Foraging capacity (m) | | <500–3,000 | 500–3,000 | >3,000 |
| Diet breadth | | Specialist–generalist | Specialist–generalist | Generalist |

● Baseline
■ Extensive foragers
▲ Limited foragers

□ Untreated seminatural
■ Contaminated seminatural
□ Treated non-flowering crop
▨ Treated flowering crop 1
▨ Treated flowering crop 2

**Fig. 1 | A trait-based, spatially explicit framework for the pesticide exposure and risk of bees. a**–**d**, We describe three sets of foraging traits of bees (based on refs. [23,30,82,83])–'extensive', 'intermediate' and 'limited' (**a**), in relation to landscape context (**b**), as demonstrated in low-intensity (**c**) and high-intensity (**d**) landscapes, whereby extensive (grey square) and limited (grey triangle) foragers move between habitat types within their respective foraging ranges (concentric circles relative to X, the central nests). Our baseline assumption (**b**, black circles) is that pesticide exposure and risk will increase with agricultural intensification, proportional to the area of agricultural land within the foraging range of bees (**c** and **d**, concentric circles). We expect bees with the largest foraging range, 'extensive' foragers, to receive the highest pesticide exposure and risk independent of landscape context (**b**, line intercept; **c** and **d**, grey squares). However, as agriculture intensifies, the proportion of agricultural land within the foraging range of bees increases and the likelihood of foraging on contaminated food increases. Therefore, we expect 'limited' foragers to be disproportionately more at risk from pesticide exposure as agricultural land expands (**b**, line slope; **c** and **d**, grey triangles). NA, not applicable.

resources and are thus more reliant on seminatural habitats to provide continuous forage. Therefore, limited foragers may be less exposed if seminatural habitats are available and provide non-contaminated forage (compare ref. [31]). However, limited foragers may become disproportionately more exposed in intensively managed agricultural landscapes, where there is an increased likelihood of contamination in the few seminatural habitats (Fig. 1b; line slope).

To test whether foraging traits alter exposure and risk for bees in different landscape contexts, we assayed pesticide residues in pollen and nectar collected by *A. mellifera*, *B. terrestris* and *O. bicornis*, representing extensive, intermediate and limited foragers, respectively, across three sequentially blooming crops (Figs. 1 and 2). In doing so, we integrate multiple domains of pesticide exposure usually restricted to single studies: landscape context (for example, ref. [32]), pollinator species (for example, ref. [33]), crops (for example, ref. [15]) and food sources (for example, ref. [34]). We predicted that pesticide exposure and risk would increase with (1) the proportion of agricultural land and (2) the extent of foraging traits. Furthermore, we expected (3) limited foragers to experience greater pesticide exposure and risk than more extensive foragers with an increasing proportion of agricultural land. Additionally, we expected (4) that mass-flowering crops were the primary source of pesticide exposure, particularly for extensive foragers and that there may be crop-specific risks based on crop-specific pest management recommendations (Supplementary Table 1). Finally, we expected (5) pesticide exposure and risk to correlate between the pollen and nectar loads of bees, with potential application to postapproval pesticide monitoring. With expected drastic changes to pesticide regulation to meet current sustainability goals (for example, ref. [35]) and calls for environmental risk assessment to become more accurate, reliable and holistic[36], it is essential to understand why different cropping patterns and landscape contexts may differentially put key pollinator species at risk.

## Results

Across bee species (*A. mellifera*, *B. terrestris* and *O. bicornis*) and crops (oilseed rape, apple and clover) for both food sources (pollen and nectar), a total of 53 compounds were detected (of the 120 screened), including 24 fungicides, 19 herbicides, 5 insecticides, 2 acaricides, 2 metabolites of herbicides and 1 metabolite of a fungicide. We detected more compounds in pollen samples from oilseed rape sites (42, $n = 40$) than apple (36, $n = 36$) and clover sites (25, $n = 32$). The four compounds with the greatest compound-specific risk were insecticides (Table 1) but some herbicides and fungicides also ranked highly due to their high concentration or frequency (Supplementary Table 2). Herbicides and fungicides comprised 80% of total detections and 65% of total residues (in µg kg$^{-1}$), yet unsurprisingly insecticides represented most of the pesticide risk, accounting for over 99% of the compound-specific risk (Supplementary Table 2).

Pesticide risk (additive toxicity-weighted concentrations; Methods) was explained by the focal crop ($F_{2,20.48} = 8.4$, $P < 0.01$) and an interaction between bee species and the proportion of agricultural land in the landscape (Fig. 3a; $R^2$m = 0.39, $F_{2,34.472} = 4.4$, $P = 0.02$) but not by an interaction between bee species and the focal crop ($F_{3,28.196} = 0.1$, $P = 0.97$) or the three-way interaction ($F_{3,28.10} = 2.3$, $P = 0.10$). The risk increased with the proportion of agricultural land for *O. bicornis* (trend estimate (confidence interval) 7.77 (2.53, 13.01)) and *B. terrestris* (7.00 (1.92, 12.08)), while that of *A. mellifera* (2.79 (−2.25, 7.83)) was independent of the proportion of agricultural land. The increase in risk was similar between *O. bicornis* and *B. terrestris* (Tukey-adjusted difference in slopes $P = 0.91$) but was stronger for *O. bicornis* than for *A. mellifera*

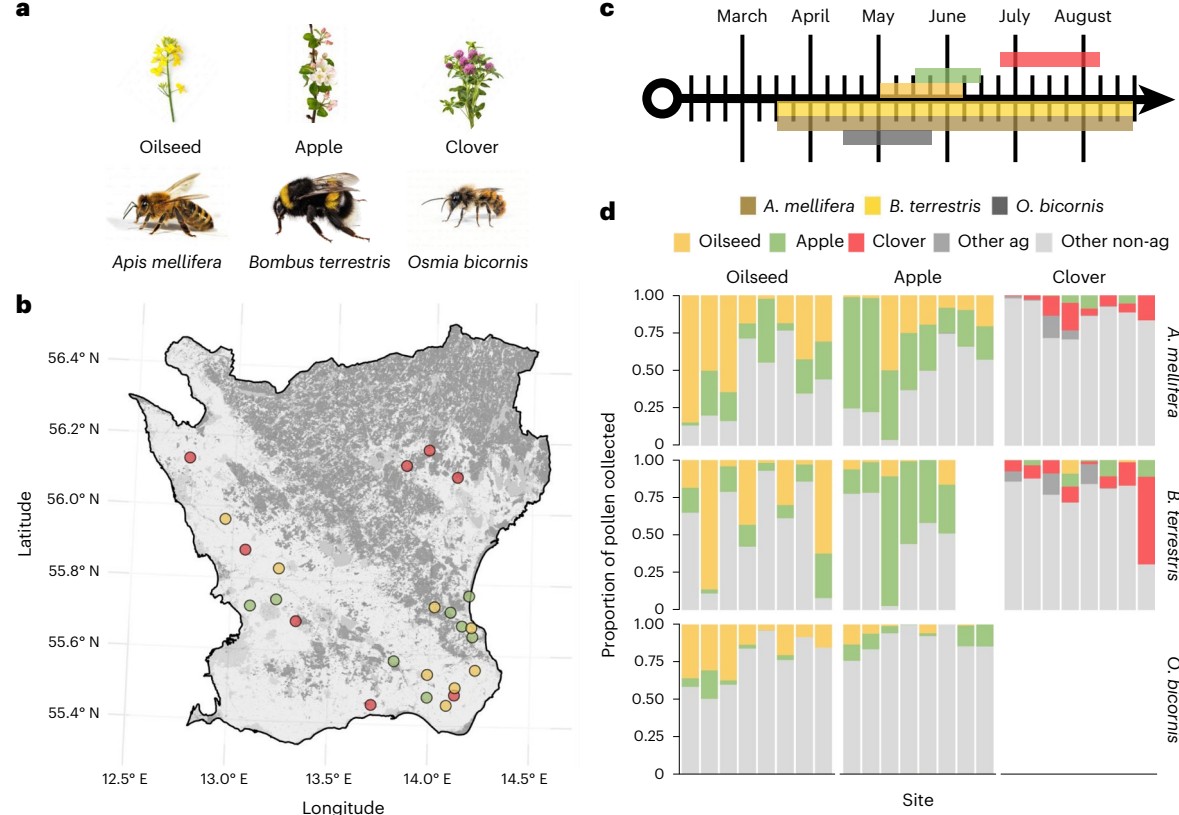

**Fig. 2 | Study design for assessing bee pesticide exposure and risk in relation to different ecological traits and landscape contexts. a,b,** We introduced sentinels of three bee species that vary in their sociality and foraging range to fields of three pollinator-dependent crops (**a**) across a gradient of land use in southernmost Sweden (**b**). Our focal bee species were *A. mellifera*, an extensive forager; *B. terrestris*, an intermediate forager; and *O. bicornis*, a limited forager. **c,** The activity periods and flowering phenology of bees and crops overlapped, except for red clover and *O. bicornis*. **d,** Non-agricultural (other non-ag) plant species/groups often dominated pollen use at each site (*x* axis) and bees tended to use more of the focal crop pollen than other agricultural (other ag) types. Pollen use and pesticide residue data are unavailable for red clover and *O. bicornis* due to non-overlapping phenologies (**c**). Due to colony failure, data are also absent for *B. terrestris* colonies at two apple sites. Images in **a** and map in **b** are free to use under creative commons licences (CC-BY and CC0).

### Table 1 | Compound-specific pesticide risk overall and for each bee species on the basis of the relevant detection rate and concentration (the latter is not shown for each species)

| Pesticide (Type) | Pesticide group | $LD_{50}$ mean | Overall | | | | *Apis mellifera* | | *Bombus terrestris* | | *Osmia bicornis* | |
|---|---|---|---|---|---|---|---|---|---|---|---|---|
| | | | Concentration (90th) | Detections | Compound-specific risk | | Detections | Compound-specific risk | Detections | Compound-specific risk | Detections | Compound-specific risk |
| Indoxacarb (I) | Oxadiazine | 0.156 | 356 | 21 (34%) | 775 | | 9 (38%) | 738 | 6 (27%) | 1,070 | 6 (38%) | 287 |
| Imidacloprid (I) | Neonicotinoid | 0.0420 | 4.50 | 3 (5%) | 5.32 | | 0 (0%) | | 1 (5%) | 0.342 | 2 (12%) | 14.0 |
| Acetamiprid (I) | Neonicotinoid | 11.3 | 65.5 | 47 (76%) | 4.40 | | 18 (75%) | 2.99 | 14 (64%) | 4.30 | 15 (94%) | 5.02 |
| Thiacloprid (I) | Neonicotinoid | 28.1 | 66.0 | 56 (90%) | 2.12 | | 24 (100%) | 2.82 | 20 (91%) | 2.04 | 12 (75%) | 3.09 |
| Metamitron (H) | Triazinone | 98.6[a] | 36.9 | 44 (71%) | 0.266 | | 15 (62%) | 0.113 | 14 (64%) | 0.709 | 15 (94%) | 0.261 |
| Penconazole (F) | Triazole | 7.10[a] | 16.4 | 6 (10%) | 0.231 | | 3 (12%) | 0.223 | 1 (5%) | 0.167 | 2 (12%) | 0.191 |
| Tebuconazole (F) | Triazole | 142[a] | 117 | 16 (26%) | 0.214 | | 11 (46%) | 0.0550 | 4 (18%) | 4.57 | 1 (6%) | 0.00200 |
| Tau-fluvalinate (I) | Pyrethroid | 12.3 | 14.0 | 4 (6%) | 0.0680 | | 4 (17%) | 0.193 | 0 (0%) | | 0 (0%) | |

[a]$LD_{50}$ based on limit tests[75]. Pesticide identity, type (I, insecticide; F, fungicide; H, herbicide; N, nematicide), group, toxicity (average acute and contact $LD_{50}$ for *A. mellifera* adults, μg per bee[71]), concentration (90th percentile, μg kg$^{-1}$), frequency of detection and compound-specific risk (Methods) of the five riskiest compounds for each species on the basis of their collected pollen and nectar.

($P$ = 0.03). The proportion of focal cropland ($F_{2,34.15}$ = 1, $P$ = 0.39) and mean-field size ($F_{2,34.35}$ = 1.04, $P$ = 0.36) in the 2 km radius landscape were not predictors of risk for any bee species.

The proportion of agricultural pollen collected by bees was also explained by focal crop ($F_{2,21.64}$ = 9, $P$ < 0.01) and an interaction between bee species and the proportion of agricultural land (Fig. 3b; $R^2$m = 0.44,

$F_{2,35.72}$ = 4.4, $P$ = 0.02), without an interaction between bee species and focal crop ($F_{3,28.70}$ = 1.99, $P$ = 0.14) or the three-way interaction ($F_{3,28.41}$ = 1.35, $P$ = 0.27). Agricultural pollen use by *O. bicornis* increased with the proportion of agriculture in the landscape (trend estimate 2.71 (0.55, 4.86)) but not for *A. mellifera* (0.01 (−1.93, 1.96)) or *B. terrestris* (−0.88 (−2.87, 1.12)). On average, bees collected 30% oilseed rape-type

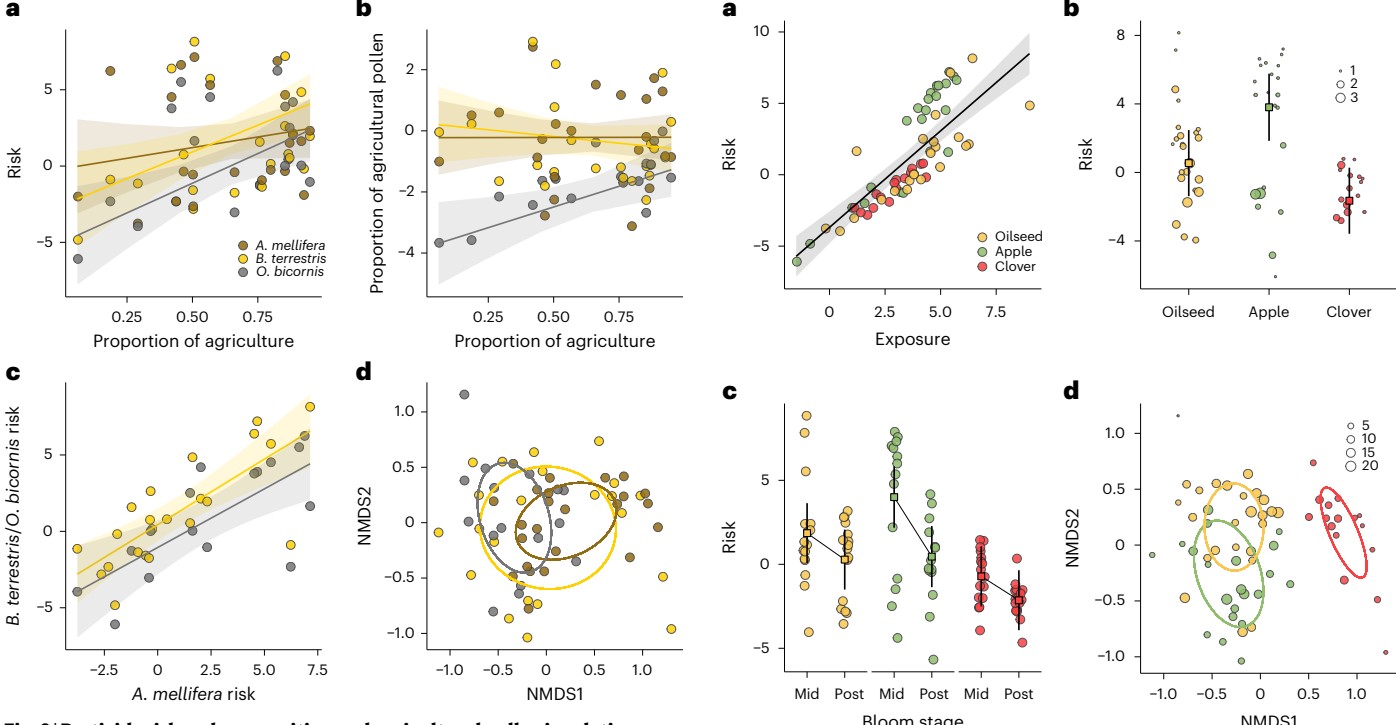

**Fig. 3 | Pesticide risk and composition and agricultural pollen in relation to bee species and landscape context. a**, Results showed that pollen-based pesticide risk increased with the amount of agricultural land in the landscape for *B. terrestris* and *O. bicornis*, while *A. mellifera* pollen-based risk was independent of agricultural land extent. **b**, The proportion of agricultural land also influenced pollen use, with only *O. bicornis* using more agricultural pollen with increasing agricultural land. **c**, Risk for *A. mellifera* correlated with that of *O. bicornis* (grey) and *B. terrestris* (yellow). **d**, The composition of pesticide compounds in pollen differed between *A. mellifera* and *O. bicornis*, while *B. terrestris* overlapped the two based on PERMANOVA of Bray–Curtis dissimilarities. Dispersion varied between bee species (*P* = 0.03); therefore, these community differences should be interpreted cautiously. Predictions and 95% confidence intervals (**a**,**b**,**c**) come from linear models with risk log transformed and the proportion agricultural pollen logit transformed. NMDS points (**d**) are based on standardised Bray–Curtis distances.

**Fig. 4 | Pesticide risk and composition in relation to focal crop and bloom stage. a**, Results show that pesticide risk and exposure were correlated (*R*² = 0.74). **b–d**, Pollen-based risk (**b**), risk relative to the timing of focal crop bloom (**c**) and the composition of pesticide compounds (**d**) differed among focal crops on the basis of PERMANOVA of Bray–Curtis dissimilarities. We scaled points in **b** by their respective MCR (Methods), to depict pesticide mixture risk relative to its constituent single most risky compound. A value close to one indicates that a single compound dominates the mixture risk. MCR values did not differ between crops (Supplementary Fig. 1). Outlined squares (**b** and **c**) depict means and 95% confidence intervals (**b**, oilseed rape *n* = 24, apple *n* = 22 and clover *n* = 16; **d**, oilseed rape *n* = 32, apple *n* = 28 and clover *n* = 32). We scaled points in **d** by the number of pesticides detected in a pollen sample. Predictions and 95% confidence intervals (**a**,**b**,**c**) come from linear models with risk log transformed. NMDS points (**d**) are based on standardised Bray–Curtis distances.

pollen at oilseed rape sites, 29% apple-type pollen at apple sites and 12% clover-type pollen at red clover sites (Fig. 2d). The proportion of focal crop in the landscape did not influence the use of focal crop pollen by bees ($F_{2,35.01}$ = 1.35, *P* = 0.27). Pesticide risk did not increase with the proportion of agricultural ($F_{2,35.28}$ = 1.13, *P* = 0.33) or focal crop pollen ($F_{2,35.64}$ = 1.40, *P* = 0.26).

We found that bee species experienced similar site-level risk— *A. mellifera* related to *B. terrestris* (Fig. 3c; *R*² = 0.6, *T* = 4.19, d.f. = 18, *P* < 0.01) and *O. bicornis* (Fig. 3c; *R*² = 0.53, *T* = 3.57, d.f. = 13, *P* < 0.01) and *O. bicornis* related to *B. terrestris* (*R*² = 0.65, *T* = 4.48, d.f. = 11, *P* < 0.01). Pesticide risk and exposure were correlated (Fig. 4a; *R*²m = 0.74, $F_{1,55.92}$ = 111.31, *P* < 0.01) and we provide parallel exposure results (additive concentrations) in the Supplementary Results.

Pollen collected at apple sites had higher risk compared to clover sites (Fig. 4b; *T* = 4.09, d.f. = 21.2, *P* < 0.01) but was similar between oilseed rape and apple sites (*T* = −2.39, d.f. = 19.5, *P* = 0.07) and oilseed rape and clover sites (*T* = 1.69, d.f. = 20.8, *P* = 0.23) (Fig. 4b). Risk (Fig. 4c) and exposure (Supplementary Fig. 2) were higher during crop bloom than after crop bloom.

Compound composition in pollen differed between the focal crops (PERMANOVA $F_{2,61}$ = 11.34, *P* < 0.01) and between bee species (PERMANOVA $F_{2,61}$ = 2.12, *P* = 0.01), without an interaction between bee

species and focal crop (*P* > 0.05). Between bee species, the compound composition only differed between *O. bicornis* and *A. mellifera* (Fig. 3d and Supplementary Table 3; $F_{1,38}$ = 3.85, *P* < 0.01). Between focal crops, all pairwise comparisons indicated different compound compositions (Fig. 4d and Supplementary Table 3, all *P* < 0.01).

Risk, not accounting for assumptions of residue intake, for example, via consumption, was higher in pollen than in nectar (Fig. 5a; *T* = −10.66, d.f. = 93.9, *P* < 0.01) and the pesticide composition differed between these sample materials (Supplementary Fig. 3, PERMANOVA $F_{1,49}$ = 2.42, *P* = 0.04). We found that the pollen-based risk related to the nectar-based risk (Fig. 5b; *R*²m = 0.10, *T* = 2.15, d.f. = 53.99, *P* = 0.04).

## Discussion

The pesticide exposure of bees arises from their activity intersecting pesticide use[12]. Thus, pesticide exposure and its correlated risk (additive toxicity-weighted concentrations) to bees are likely to be affected by their life-history traits[37], particularly foraging habits[23,26,38] and land-use and pesticide-use patterns, especially in bee-attractive crops[39,40]. Using an ecological approach to pesticide risk, we found that extensive foragers (*A. mellifera*) experienced the greatest risk irrespective of the proportion of agricultural land in the landscape. Although risk correlated among bee species, both limited foragers (*O. bicornis*)

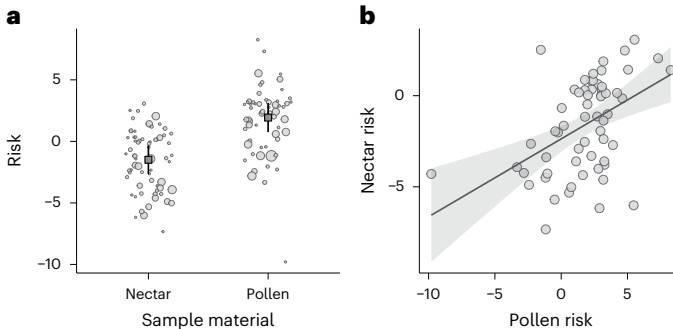

**Fig. 5 | Pesticide risk in bee-collected pollen and nectar. a**, Results show that pesticide risk was greater from pollen than nectar, but the risk correlated between sample materials. As in Fig. 4b, points in **a** are scaled by their respective MCR, where a smaller point indicates that a single compound dominates the pesticide mixture risk. Outlined squares (**a**) depict mean log transformed risk (nectar $n = 70$ and pollen $n = 61$). Predictions and 95% confidence intervals (**a,b**) come from linear mixed-effects models with risk log transformed. **b**, We re-analysed data with the left-hand outlier removed and the results were qualitatively unchanged and the model fit improved.

and intermediate foragers (*B. terrestris*) experienced less risk than extensive foragers (*A. mellifera*) in landscapes with less agricultural land. In addition, risk correlated between sample materials and was greatest in pollen. Consequently, *A. mellifera*-collected pollen can cautiously predict pesticide risk for bees, not accounting for residue intake, compared to nectar and pollen collected by other bee species, independent of landscape context. Thus, the *A. mellifera*-collected pollen-based pesticide risk indicator may be a promising metric for postapproval pesticide monitoring in terrestrial systems, generally proposed by ref. [41] and with parallels in aquatic systems[42].

Agricultural landscapes expose *A. mellifera* to multiple pesticides[15,27,32,43–45]. However, we know less about the resulting pesticide risk, especially between bee species and in different landscape contexts (but see refs. [22,26,46,47]). We found that increasing the proportion of agricultural land increased the risk for *B. terrestris* and *O. bicornis* but not for *A. mellifera*. We suggest that these landscape-dependent differences in risk result from species-specific activity patterns[23,38]. Different crop pollen use between the three species somewhat supports this: uniform collection by *A. mellifera* and *B. terrestris* compared to increasing collection by *O. bicornis* with an increasing proportion of agricultural land, consistent with findings in apple for *A. mellifera*[32] and *O. cornifrons*[28]. Consequently, mass-flowering crops appear to be a predominant food source for *A. mellifera* across agricultural landscapes[32,38,40,45]. In contrast, despite access to mass-flowering crops, *O. bicornis* favours non-crop, predominantly woody, pollen resources when available[48,49]. These different preferences for crop pollen are evidenced by others finding that the collection of focal crop pollen positively correlates to the proportion of that crop in the landscape for *A. mellifera* (apple[32]) and *B. impatiens* (blueberry[27]) but not *O. bicornis* (oilseed rape[31,48]). Therefore, sets of foraging traits (for example, large colony size and advanced communication) and foraging preferences probably drive the prevalence of *A. mellifera* in mass-flowering crops. In intensively managed agricultural landscapes with scarce seminatural habitats and high pesticide use, *O. bicornis* is increasingly likely to forage in less-preferred mass-flowering crops and seminatural habitats adjacent to arable land[31] and thus increase their pesticide exposure and risk. Consequently, populations of *O. bicornis* and similar, limited foragers may be disproportionately affected by agricultural intensification as their traits compound the combined effects of habitat loss and increased pesticide exposure[26]. Our use of *O. bicornis* as a sentinel allowed us to estimate exposure and risk of a limited forager

in landscapes where they may not naturally occur, which, combined with the relatively generalised diet of *Osmia* spp.[26,48,50], means that our estimates for limited foragers are probably precautionary among solitary bee species.

The focal crop (oilseed rape, apple or clover) was an important driver of pollen-derived exposure and risk for all bee species, independent of the proportion of agricultural land. For example, all bee species experienced the highest exposure and risk at apple sites, followed by oilseed rape and clover sites. These results mirror the approved number of active ingredients in plant protection products recommended for use in the three focal crops, with most in apple and least in clover. Apple and other fruit crops generally have higher pesticide use[51] and resulting bee exposure than annual arable crops or permanent grasslands[15]. We also found that the composition of pesticides in pollen differed between the three crops, identifying pest management strategies for specific crops and even specific compounds as determinants of landscape-level exposure and risk. Pollen pesticide risk was greater during crop bloom than after bloom across all three investigated crops. However, it did not correlate with either agricultural or focal crop pollen collection, possibly pointing toward the treated crop and associated flowering plants affected by drift as sources of pesticide exposure[20,22,26,27]. Focusing on spatiotemporally matched pollen and nectar samples from *A. mellifera* and *B. terrestris*, we found that exposure and risk were higher in pollen than in nectar, although this does not account for the uptake of residues by bees for example via consumption which is unequal between pollen and nectar[33]. Nonetheless, we found that risk but not exposure positively correlated between pollen and nectar; thus, pollen may be a precautionary material for estimating the pesticide risk of bees and, more generally, pesticide contamination of terrestrial environments[34,52].

Pollen pesticide mixture composition differed the most between *A. mellifera* and *O. bicornis*, while *B. terrestris* overlapped the two. The three species shared two of the riskiest compounds, indoxacarb and acetamiprid, while the following most risky compounds were unique to each species: thiacloprid for *A. mellifera*, tebuconazole for *B. terrestris* and imidacloprid for *O. bicornis*. Nevertheless, risk positively correlated among the three species, suggesting that risk estimates for one species can, to some degree, inform the risk to other bee species. The generally low maximum cumulative ratio (MCR) values indicate that the pesticide mixture risk, independent of bee species and focal crop, was driven by one or a few high-risk compounds (similar to ref. [53]). High-risk compounds were mainly neonicotinoid insecticides (acetamiprid, imidacloprid and thiacloprid), previously identified as high-risk to bees[33,54] but the riskiest compound was indoxacarb, an oxadiazine insecticide. Reduced exposure to these high-risk compounds would substantially decrease the risk for the three bee species. In the EU, pesticide restrictions (imidacloprid 2018, thiacloprid 2021 and indoxacarb 2022) are regulatory moves in this direction[55–57], even if residues persist (like imidacloprid in our study[58]) or new compounds with similar risk profiles enter the market in the future[59,60].

Pesticide risk assessment primarily focuses on *A. mellifera*, partly because of its economic value, ease of management and a greater understanding of the species' biology[61–63]. However, risk assessment is becoming more holistic[36], with a greater emphasis on non-*Apis* species[64] in recognition of wild bee diversity and their contribution to pollination services[65]. However, this change requires a better understanding of how pesticide risk varies among bee species and landscape contexts. We found that the pesticide risk estimated from *A. mellifera*-collected pollen was generally higher than or similar to *B. terrestris* and *O. bicornis*, particularly in landscapes with less agricultural land. Thus, whilst bee traits regulate pesticide exposure and risk, there is potential to extrapolate risk among bee species and exposure sources, with higher and thus precautionary risk estimates based on *A. mellifera*-collected pollen. However, pesticide exposure and our ecological indicator of pesticide risk do not account for species-specific

processes past the pesticide use–bee activity intersection, such as consumption within the nest or indirect effects that could affect the fitness of the bees–important considerations when moving from exposure to effect in environmental risk assessment[63].

Using our trait-based approach, we conclude that landscape context modifies pesticide risk but only for limited and intermediate foragers (here, *O. bicornis* and *B. terrestris*, respectively). These findings highlight the potential for seminatural habitats to buffer pesticide-related risks for wild bees[26,46,66]. We also conclude that *A. mellifera*-collected pollen can predict environmental pesticide risk for other species and is precautionary, particularly in less agriculturally dominated landscapes. We, therefore, suggest that an *A.* mellifera-collected pollen-based pesticide risk indicator is a promising metric for postapproval pesticide monitoring in terrestrial systems (compare ref. [41]). However, questions remain as to how this exposure affects individuals and, ultimately, populations of bees–tasks for a more holistic and realistic environmental risk assessment that aims to capture exposure to pesticide mixtures and risks within the diverse bee community[67].

## Methods

### Field site system and sentinel bees

We centred 24 sites on three bee-attractive flowering crops: oilseed rape (8 sites), apple (8 sites) and red clover grown for seed production (8 sites) in southern Sweden (Fig. 2). These crops bloom sequentially: oilseed rape during April–May, apple during May–June and red clover during June–August (Fig. 2c) and are affected by different pests and therefore have different pest management strategies. The national pest management recommendations for 2019 included 26 active ingredients in oilseed rape, 32 in apple and 14 in clover seed and included acaricide (2 active ingredients), fungicide (20), herbicide (20) and insecticide (13) products (Supplementary Table 1). We selected sites on the basis of their surrounding proportion of agricultural land (2 km radius) to ensure an even gradient (for each crop type) of agricultural land and, therefore, anticipated pesticide use[15,16,68]. The average (± s.d.) proportion of agricultural land was 74 ± 24% (range 29–95%) for oilseed rape, 52 ± 29% (6–85%) for apple and 66 ± 20% (44–93%) for clover. All sites were >6 km apart, except for two clover sites, 2 km apart. Southern Sweden is characterised by annual crop production and nationally high pesticide use[69]. Farmers managed crops conventionally, except for one field of each focal crop, which was managed organically.

In 2019, we placed sentinel bees at focal crop fields at the onset of flowering and allowed them to forage freely without supplemental food. At each field, we placed: (1) two or three nationally produced, standardised and conventionally managed *A. mellifera* colonies, (2) six commercial *B. terrestris* colonies (Biobest Biological Systems) in two large ventilated wooden boxes and (3) three solitary bee trap nest units (at the oilseed rape and apple sites) each seeded with 50 male and 50 female *O. bicornis* cocoons (Wildbiene & Partner) (Supplementary Methods). We did not place *O. bicornis* in clover fields as their phenologies do not overlap (Fig. 2c).

### Quantification of pesticide residues in pollen and nectar

We sampled pollen from (1) *A. mellifera* using pollen traps attached to two hives for 24 h, (2) *B. terrestris* by capturing foragers (~20 across all six colonies) and killing them on dry ice as they returned to their colonies and (3) multiple *O. bicornis* brood cell pollen provisions collected by females over the second half of the bloom period. We sampled pollen from *A. mellifera* and *B. terrestris* at two sampling intervals, coinciding with (1) the peak of crop bloom and (2) after crop bloom and for *O. bicornis* only at the end of crop bloom (evenly from all the available pollen). In total, we collected 48 samples (595 g) of *A. mellifera*-collected pollen, 44 samples (11 g) of *B. terrestris*-collected pollen and 16 samples (70 g) of *O. bicornis*-collected pollen. During and after bloom, samples were pooled for both *A. mellifera* and *B. terrestris*, resulting in 24 samples of *A. mellifera*-collected pollen, 22 samples of

*B. terrestris*-collected pollen (all colonies died at two sites) and 16 samples of *O. bicornis*-collected pollen. We did not pool *O. bicornis* pollen over the bloom period since this species already combined pollen provisions on our behalf.

To compare residues between nectar and pollen, we sampled additional returning foragers of *A. mellifera* ($n \approx 100$ individuals per sample) and *B. terrestris* ($n \approx 20$ individuals per sample) 1–2, 4–6 and 12–16 days after a known pesticide application at four oilseed rape, two apple and seven clover sites (Supplementary Table 4). Corbicular pollen and nectar stomach content were collected from these foragers to produce paired pollen and nectar samples for each site and collection time point ($n = 54$).

We froze pollen and bee samples, before nectar extraction, at −20 °C before screening for 120 pesticide compounds included in the Swedish national monitoring scheme (Supplementary Table 5), following established protocols at the Laboratory for Organic Environmental Chemistry (SLU) (Supplementary Methods).

### Pollen identification

Part of each pollen sample was analysed to determine the pollen use of the three bee species at each site. First, we pooled pollen samples per site, bee species and bloom period in a 5 ml tube and agitated them in 5 ml of 70% ethanol before pipetting 2 µl of the pollen suspension onto a microscope slide stained and set using fuchsin gel under a coverslip. Next, we identified (using a pollen reference library at the Department of Biology (Lund) and ref. [70]) and counted >400 pollen grains per slide (7–20 rows, 163 µm wide across the slide) using ×400 magnification. On the basis of this, we quantified the proportional use of all agricultural-type pollen and focal crop pollen by bees and categorised the latter into a Brassicacae group (including oilseed rape; *Brassica napus*), *Malus* group (including apple; *Malus domestica*) and *Trifolium pratense* group (including red clover; *T. pratense*) (Supplementary Table 6).

### Landscape classification

We analysed the landscape surrounding our sites at multiple spatial scales (1,000, 1,500 and 2,000 m, corresponding to the average foraging capacities of bees (Fig. 1a)) on the basis of the IACS Spatial Data Layer provided by the Swedish Board of Agriculture. We classified land cover categories into two groups: agricultural land (all types of agricultural use, such as annual crops, orchards, leys and seminatural grasslands) and non-agricultural land (including forest, urban areas and water bodies). This distinction is because our focus was on the pesticide exposure and risk to bees from agricultural pesticide use and the pesticide exposure of bees is higher in rural compared to urban areas[22]. We also calculated the proportion of the focal crop in the radii and the average field size. We confirmed that the proportion of agricultural land was consistent (Supplementary Fig. 8) and correlated (Supplementary Fig. 9) across the three spatial scales for each crop type and consequently used the landscape information at the largest scale (2,000 m) in all subsequent analyses.

### Risk calculations

We use toxicity-weighted concentrations (TWC) as a basis for indicating pesticide risk for bees[26], where the TWC of each compound ($TWC_i$) is the ratio between the concentration ($c_i$) of a detected compound in bee-collected pollen or nectar and its respective acute toxicity endpoint ($LD_{50i}$–the dose required to cause 50% mortality in the test population)[71]. Then, following a concentration addition approach–the recommended default for mixture environmental risk assessment[72] (even though some compound classes may synergize[73]), we summed TWCs, to calculate the additive toxicity-weighted concentration of all compounds within a sample per site and bee species ($TWC_{mix}$):

$$TWC_{mix} = \sum_{i=1}^{n} \frac{c_i}{LD_{50}i}$$

Henceforth, we refer to this metric, an indicator of pesticide-related risk, as 'risk'.

We averaged the acute oral and contact $LD_{50}$ (ref. [71]) of each compound to provide an overall indicator of toxicity, reflective of how bees encounter pesticides in the landscape and their multiple exposure routes[37]. We used $LD_{50}$ for adult *A. mellifera* because there are incomplete toxicity data for other bee species and life stages and, where there are data, $LD_{50}$ for other bee species correlate with the corresponding *A. mellifera* $LD_{50}$ (refs. [53,74]). Furthermore, in using the same $LD_{50}$ across bee species, we disentangle the ecology of bees from toxicology to explore relative differences in the activity patterns of bees in intersection with pesticide use. Finally, we used the tested dose for $LD_{50}$ based on limit tests[71] (used when a compound is expected to be low in toxicity or there are issues with solubility[75]), which can overestimate the toxicity of a compound. Three of these compounds ranked highly for compound-specific risk due to their high concentrations and frequency of detection rather than toxicity (Table 1).

We also calculated the factor by which the mixture risk ($TWC_{mix}$) was greater than its composite most risky compound ($max(TWC_i)$) using an MCR[76]. Thus, an MCR close to one indicates that a single compound dominates risk. The MCR did not vary among bee species or focal crops (Supplementary Fig. 1).

Finally, we also calculated compound-specific risk (Table 1 and Supplementary Table 2) to identify high-risk compounds by multiplying $TWC_i$ by its bee-specific detection frequency[33].

## Data analyses

We conducted four primary analyses to understand agricultural pesticide risk to bee species, followed by supporting multivariate analyses of the compound compositions. We performed analyses and data visualization using R v.4.1.1., constructed linear mixed-effects models (LMMs) with the lme4 package[77] and analysed compound composition with the vegan package[78]. For the primary analyses, risk data were log transformed and the proportion of crop pollen was logit transformed to meet assumptions of normality and homogeneity of variance. Upon detecting significant main effects, we examined the significance and difference of individual factor levels via pairwise comparisons of estimated marginal means using Tukey's method with the emmeans package[79]. Finally, we evaluated models for overdispersion and checked residuals for normality and homoscedasticity using diagnostic functions in the performance package[80]. We report marginal $R^2$ values calculated following the methods of ref. [81].

**Risk and pollen use with landscape context, focal crop and bee species.** We used LMMs to explore (1) risk from pollen and (2) use of agricultural pollen, with focal crop and bee species interacting with the proportion of agricultural land as fixed effects and site as a random intercept. We included an interaction between bee species and crop for both analyses but this was non-significant and thus removed. Additionally, we used a similar model, including the focal crop, bee species interaction and site as random intercept, to relate focal crop pollen to the proportion of that focal crop in the landscape.

**Risk with sampling round and focal crop.** We tested whether risk varied between the different sampling rounds using an LMM with sample round, focal crop and bee species included as fixed effects and site as a random intercept. Finally, we tested if risk related to the proportion of focal crop pollen, bee species and focal crop, with focal crop pollen interacting with bee species, as fixed effects and site as a random intercept.

**Risk among bee species.** We examined risk relationships among the site-specific pollen collection of bee species using three linear models, one for each species. We included the remaining bee species and focal crop as fixed effects; however, the focal crop was non-significant in all models ($P > 0.05$).

**Risk between sample materials.** We used data from the paired pollen–nectar collections to test for a difference in risk between sample materials (pollen versus nectar), using LMMs with sample material, focal crop and bee species as fixed effects, and sampling round nested in the site as a random intercept. In addition, we examined risk relationships among sample material collections, using an LMM with nectar risk specified as the response variable and pollen risk, focal crop and bee species as fixed effects, and sampling round nested in the site as a random intercept.

**Differences in compound composition.** We used PERMANOVA to compare the composition of compounds between focal crops and bee species using a Bray–Curtis dissimilarity index based on a Hellinger standardised community matrix of risk values using the adonis2() function in vegan. We used non-metric multidimensional scaling (NMDS) to visualise different clusters of compounds. We tested for differences in dispersion between focal crops or bee species using the betadisper() function in vegan. We detected no differences in the dispersion of compounds between crops. However, we found different dispersion of compounds between bee species ($P = 0.03$); therefore, we should interpret these community differences cautiously.

## Reporting summary

Further information on research design is available in the Nature Portfolio Reporting Summary linked to this article.

## Data availability

Data available via Figshare https://doi.org/10.6084/m9.figshare.20390751.

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

## Acknowledgements

We thank the farmers and landowners for providing access to their land, beekeepers for colony management, A. Bates, O. Ronsevych, G. Svensson, E. T. Talavan and T. Krausl for sample collection and handling, M. Karlsson for pollen identification and D. Sponsler and C. Stuligross for their constructive and thought-provoking feedback on the manuscript. This research was supported by the Swedish research council Formas (2018-02283 (M.R. and O.J.)), 2018-01020 (M.R. and J.R.d.M.)), the strategic research area BECC (Biodiversity and Ecosystem Services in a Changing Climate; 2016/1873 (M.R.)) and the European Union Horizon 2020 PoshBee (Pan-European assessment, monitoring and mitigation of stressors on the health of bees; 773921 (J.R.d.M.)).

## Author contributions

J.L.K. made substantial contributions to conception, acquisition and interpretation of data and drafted the article. C.C.N. made substantial contributions to conception, analysis and interpretation of data and drafted the article. O.J. was involved in acquisition of funding and acquisition, analysis and interpretation of data. J.R.d.M helped with acquisition of funding and data. M.R. contributed to acquisition of funding, made substantial contributions to conception and design, acquisition and interpretation of data and drafted the article. All authors critically revised the paper for important intellectual content.

## Funding

## Competing interests

The authors declare no competing interests.

## Additional information

**Correspondence and requests for materials** should be addressed to Jessica L. Knapp or Maj Rundlöf.

# Reporting Summary

## Statistics

For all statistical analyses, confirm that the following items are present in the figure legend, table legend, main text, or Methods section.

| n/a | Confirmed | |
|---|---|---|
| ☐ | ☒ | The exact sample size ($n$) for each experimental group/condition, given as a discrete number and unit of measurement |
| ☐ | ☒ | A statement on whether measurements were taken from distinct samples or whether the same sample was measured repeatedly |
| ☐ | ☒ | The statistical test(s) used AND whether they are one- or two-sided <br> *Only common tests should be described solely by name; describe more complex techniques in the Methods section.* |
| ☐ | ☒ | A description of all covariates tested |
| ☐ | ☒ | A description of any assumptions or corrections, such as tests of normality and adjustment for multiple comparisons |
| ☐ | ☒ | A full description of the statistical parameters including central tendency (e.g. means) or other basic estimates (e.g. regression coefficient) AND variation (e.g. standard deviation) or associated estimates of uncertainty (e.g. confidence intervals) |
| ☐ | ☒ | For null hypothesis testing, the test statistic (e.g. $F$, $t$, $r$) with confidence intervals, effect sizes, degrees of freedom and $P$ value noted <br> *Give P values as exact values whenever suitable.* |
| ☒ | ☐ | For Bayesian analysis, information on the choice of priors and Markov chain Monte Carlo settings |
| ☒ | ☐ | For hierarchical and complex designs, identification of the appropriate level for tests and full reporting of outcomes |
| ☒ | ☐ | Estimates of effect sizes (e.g. Cohen's $d$, Pearson's $r$), indicating how they were calculated |

*Our web collection on statistics for biologists contains articles on many of the points above.*

## Software and code

Policy information about availability of computer code

| Data collection | No software was used to collect data. |
|---|---|
| Data analysis | We used no custom computer code or algorithms for data analysis. All data were analysed in R version 4.1.1. |

For manuscripts utilizing custom algorithms or software that are central to the research but not yet described in published literature, software must be made available to editors and reviewers. We strongly encourage code deposition in a community repository (e.g. GitHub). See the Nature Portfolio guidelines for submitting code & software for further information.

## Data

Policy information about availability of data

All manuscripts must include a data availability statement. This statement should provide the following information, where applicable:

- Accession codes, unique identifiers, or web links for publicly available datasets
- A description of any restrictions on data availability
- For clinical datasets or third party data, please ensure that the statement adheres to our policy

The datasets generated and analysed during the current study are available from the corresponding author on reasonable request.

# Field-specific reporting

Please select the one below that is the best fit for your research. If you are not sure, read the appropriate sections before making your selection.

☐ Life sciences ☐ Behavioural & social sciences ☒ Ecological, evolutionary & environmental sciences

For a reference copy of the document with all sections, see nature.com/documents/nr-reporting-summary-flat.pdf

# Life sciences study design

All studies must disclose on these points even when the disclosure is negative.

| Sample size | Describe how sample size was determined, detailing any statistical methods used to predetermine sample size OR if no sample-size calculation was performed, describe how sample sizes were chosen and provide a rationale for why these sample sizes are sufficient. |
| --- | --- |
| Data exclusions | Describe any data exclusions. If no data were excluded from the analyses, state so OR if data were excluded, describe the exclusions and the rationale behind them, indicating whether exclusion criteria were pre-established. |
| Replication | Describe the measures taken to verify the reproducibility of the experimental findings. If all attempts at replication were successful, confirm this OR if there are any findings that were not replicated or cannot be reproduced, note this and describe why. |
| Randomization | Describe how samples/organisms/participants were allocated into experimental groups. If allocation was not random, describe how covariates were controlled OR if this is not relevant to your study, explain why. |
| Blinding | Describe whether the investigators were blinded to group allocation during data collection and/or analysis. If blinding was not possible, describe why OR explain why blinding was not relevant to your study. |

# Behavioural & social sciences study design

All studies must disclose on these points even when the disclosure is negative.

| Study description | Briefly describe the study type including whether data are quantitative, qualitative, or mixed-methods (e.g. qualitative cross-sectional, quantitative experimental, mixed-methods case study). |
| --- | --- |
| Research sample | State the research sample (e.g. Harvard university undergraduates, villagers in rural India) and provide relevant demographic information (e.g. age, sex) and indicate whether the sample is representative. Provide a rationale for the study sample chosen. For studies involving existing datasets, please describe the dataset and source. |
| Sampling strategy | Describe the sampling procedure (e.g. random, snowball, stratified, convenience). Describe the statistical methods that were used to predetermine sample size OR if no sample-size calculation was performed, describe how sample sizes were chosen and provide a rationale for why these sample sizes are sufficient. For qualitative data, please indicate whether data saturation was considered, and what criteria were used to decide that no further sampling was needed. |
| Data collection | Provide details about the data collection procedure, including the instruments or devices used to record the data (e.g. pen and paper, computer, eye tracker, video or audio equipment) whether anyone was present besides the participant(s) and the researcher, and whether the researcher was blind to experimental condition and/or the study hypothesis during data collection. |
| Timing | Indicate the start and stop dates of data collection. If there is a gap between collection periods, state the dates for each sample cohort. |
| Data exclusions | If no data were excluded from the analyses, state so OR if data were excluded, provide the exact number of exclusions and the rationale behind them, indicating whether exclusion criteria were pre-established. |
| Non-participation | State how many participants dropped out/declined participation and the reason(s) given OR provide response rate OR state that no participants dropped out/declined participation. |
| Randomization | If participants were not allocated into experimental groups, state so OR describe how participants were allocated to groups, and if allocation was not random, describe how covariates were controlled. |

# Ecological, evolutionary & environmental sciences study design

All studies must disclose on these points even when the disclosure is negative.

| Study description | We selected 24 sites centred on three bee attractive flowering crops common in the region and flowering sequentially: oilseed rape (8 sites), apple (8 sites) and red clover (8 sites). We selected sites based on their surrounding proportion of agricultural land (2km radius) to ensure an even gradient (for each crop type) of agricultural land and, therefore, anticipated pesticide use. |
| --- | --- |

Within the replicated sites, we sampled pollen from (1) Apis mellifera using pollen traps attached to two hives for 24 hours, (2) Bombus terrestris by capturing foragers (~20 across all six colonies) and euthanising them on dry ice as they returned to their colonies, and (3) multiple Osmia bicornis brood cells collected by females over the second half of the bloom period. We sampled pollen from A. mellifera and B. terrestris at two sampling intervals, coinciding with (1) the peak of crop bloom and (2) after crop bloom and for O. bicornis only at the end of crop bloom (evenly from all the available pollen). In total, we collected 48 samples (595 g) of A. mellifera-, 44 samples (11 g) of B. terrestris-, and 16 samples (70 g) of O. bicornis- collected pollen. During and after bloom samples were pooled for both A. mellifera and B. terrestris, resulting in 24 samples of A. mellifera-, 22 samples of B. terrestris- (all colonies died at two sites), and 16 samples of O. bicornis-collected pollen. We did not pool O. bicornis pollen over the bloom period since this species already combined pollen provisions on our behalf. In addition, we collected additional samples of corbicular pollen and and foraging bees for nectar extraction to produce paired pollen and nectar samples 1-2, 4-6 and 12-16 days after known pesticide applications at a selection of focal sites (n = 54 site and time point combinations for each material).

We conducted three primary analyses to understand agricultural pesticide risk to bee species. 1) Linear mixed models (LMMs) explore risk from pollen and use of agricultural pollen, with focal crop and bee species interacting with the proportion of agricultural land as fixed effects and site as a random intercept. 2) three linear models explore risk relationships among bee species' site-specific pollen collection, one for each species with the remaining bee species and focal crop as fixed effects. 3) LMMs to explore risk between sample materials, with sample material, focal crop and bee species as fixed effects, and sampling round nested in the site as a random effect.

| | |
|---|---|
| Research sample | We conducted our study in Scania, Southern Sweden, an intensively farmed European production region. The selected sites cover multiple mass-flowering crops and agricultural contexts representing temperate agricultural landscapes and different pesticide use patterns. The selected three bee species exemplify different life-history traits that determine their activity. Both these aspects are important since bees are predicted to encounter pesticides as their activity intersects pesticide use patterns. The three bee species were A. mellifera, B. terrestris and O. bicornis, representing extensive, intermediate and limited bee foragers. A. mellifera colonies came from an experienced beekeeper, free of disease signs and managed according to good beekeeping practice by professional beekeepers. B. terrestris colonies were commercially reared (Biobest Biological Systems, Belgium), free of disease signs and were placed in large ventilated wooden boxes. O. bicornis cocoons came from a commercial breeder (Wildbiene & Partner, Switzerland), free of disease signs and provided unlimited nesting tubes. |
| Sampling strategy | No pre-study sample size calculation was performed. Instead, we relied on previous experience from the region on the needed replication for pesticide exposure and risk evaluations for bees. We have previously used a replication of 6-8 sites per factor level for bee pesticide exposure studies. In this study we used a replication of 8. We selected 24 sites in total centered on three bee attractive flowering crops common in the region and flowering sequentially: oilseed rape (8 sites), apple (8 sites) and red clover (8 sites). We selected sites based on their surrounding proportion of agricultural land (2km radius) to ensure a gradient (similar for each crop type) of agricultural land and, therefore, anticipated pesticide use, similarly to how we have selected sites in previous studies.

Within these sites, we collected a sufficient amount of pollen to represent pollen plant source and for pesticide residue quantification; we collected 48 samples covering the 24 sites and 2 time points (in total 1210 g) from A. mellifera and 44 samples covering 22 sites (all colonies were lost at two apple sites which excluded sampling) and 2 time points (in total 1309 pollen loads) from B. terrestris. Sampling was restricted beyond this to exclude negatively affecting colony functioning. Furthermore, we collected all pollen possible from O. bicornis, which were 16 samples (71 g). |
| Data collection | The authors and paid research assistants collected the data and samples of bees and bee-collected pollen from April to August 2019, using pollen traps and insect nets. Sample per bee species, type of material, site and sampling time point were separately stored in tubes on ice until return to the laboratory at the end of the day, after which samples were frozen (-20C). Pollen samples and bee bodies were sent to the analytical laboratory for nectar extraction and pesticide residue analysis. Half of the sample pollen was used to identify plant species origin. Data were noted in spreadsheets on sample origin and amount of material along with pesticide identity and concentration, as well as plant species origin for pollen samples. Land use data was based on the IACS Spatial Data Layer provided by the Swedish Board of Agriculture for the study year (2019) and extracted using R. Data on pesticide properties and toxicity information were extracted from the Pesticide Properties Database (PPDB) hosted by the University of Hertfordshire. |
| Timing and spatial scale | We collected samples of pollen and bees from each focal crop field during and after bloom. Our crops bloomed sequentially: oilseed rape from April-May, apple from May-June and red clover from June-August, so that in addition to covering focal crop blooms we also covered most of the relevant season for the focal bee species. Samples at oilseed rape sites were collected 14-24 May (mid-bloom) and 22 May-7 June (end-post bloom), samples at apple sites were collected 20 May-2 June (mid-bloom) and 21 May-11 June (end-post bloom) and samples at clover sites were collected 25 June-18 July (mid-bloom) and 12-26 July (end bloom). In addition, we collected additional samples coinciding with a known pesticide application (before or during flowering) at a selection of focal sites during 14 May-25 July, all in 2019. Thus all samples were collected from May to July 2019. We conducted our study across Scania, Southern Sweden, an intensively farmed European production region covering about 100 x 100 km. |
| Data exclusions | No data were excluded. |
| Reproducibility | All data will be made freely available upon publication, and our manuscript describes our methodology in full. We have made no attempts to repeat this experiment. |
| Randomization | Sites centered on the three focal crops were selected to cover similar gradients of agricultural land and, therefore, anticipated pesticide use in the landscape. Proportion agricultural land was checked for consistency across bee relevant scales (1000, 1500, 2000 m radius). We matched honeybee hives and bumblebee colonies by their strength and randomly allocated them and Osmia cocoons to sites to ensure similar foraging efforts. |
| Blinding | Blinding during field sampling was not possible since focal bee species, focal crop and landscape context were clearly visible. |

Did the study involve field work?    ☒ Yes    ☐ No

## Field work, collection and transport

| | |
|---|---|
| Field conditions | We collected all data on warm and dry days when bees were active. |
| Location | Scania, Southern Sweden: 55.9903° N, 13.5958° E |
| Access & import/export | B. terrestris colonies were imported through a company (Lindesro AB) that held an import permit according to the Swedish Board of Agriculture. A. mellifera colonies were assessed for their health and moved to the study sites according to approval by the County Administrative Board. We needed no other permits for this study, and all farmers provided their full permission for us to access their land for fieldwork. |
| Disturbance | Disturbance was limited by removing the managed bees after the sampling was concluded. |

# Reporting for specific materials, systems and methods

We require information from authors about some types of materials, experimental systems and methods used in many studies. Here, indicate whether each material, system or method listed is relevant to your study. If you are not sure if a list item applies to your research, read the appropriate section before selecting a response.

### Materials & experimental systems

| n/a | Involved in the study |
|---|---|
| ☒ | Antibodies |
| ☒ | Eukaryotic cell lines |
| ☒ | Palaeontology and archaeology |
| ☐ | ☒ Animals and other organisms |
| ☒ | Human research participants |
| ☒ | Clinical data |
| ☒ | Dual use research of concern |

### Methods

| n/a | Involved in the study |
|---|---|
| ☒ | ChIP-seq |
| ☒ | Flow cytometry |
| ☒ | MRI-based neuroimaging |

## Antibodies

| | |
|---|---|
| Antibodies used | *Describe all antibodies used in the study; as applicable, provide supplier name, catalog number, clone name, and lot number.* |
| Validation | *Describe the validation of each primary antibody for the species and application, noting any validation statements on the manufacturer's website, relevant citations, antibody profiles in online databases, or data provided in the manuscript.* |

## Eukaryotic cell lines

Policy information about cell lines

| | |
|---|---|
| Cell line source(s) | *State the source of each cell line used.* |
| Authentication | *Describe the authentication procedures for each cell line used OR declare that none of the cell lines used were authenticated.* |
| Mycoplasma contamination | *Confirm that all cell lines tested negative for mycoplasma contamination OR describe the results of the testing for mycoplasma contamination OR declare that the cell lines were not tested for mycoplasma contamination.* |
| Commonly misidentified lines (See ICLAC register) | *Name any commonly misidentified cell lines used in the study and provide a rationale for their use.* |

## Palaeontology and Archaeology

| | |
|---|---|
| Specimen provenance | *Provide provenance information for specimens and describe permits that were obtained for the work (including the name of the issuing authority, the date of issue, and any identifying information). Permits should encompass collection and, where applicable, export.* |
| Specimen deposition | *Indicate where the specimens have been deposited to permit free access by other researchers.* |
| Dating methods | *If new dates are provided, describe how they were obtained (e.g. collection, storage, sample pretreatment and measurement), where they were obtained (i.e. lab name), the calibration program and the protocol for quality assurance OR state that no new dates are provided.* |

☐ Tick this box to confirm that the raw and calibrated dates are available in the paper or in Supplementary Information.

| | |
|---|---|
| Ethics oversight | *Identify the organization(s) that approved or provided guidance on the study protocol, OR state that no ethical approval or guidance* |

| Ethics oversight | *was required and explain why not.* |

Note that full information on the approval of the study protocol must also be provided in the manuscript.

## Animals and other organisms

Policy information about studies involving animals; ARRIVE guidelines recommended for reporting animal research

| Laboratory animals | We collected samples from three managed bee species at focal sites: Apis mellifera, Bombus terrestris, and Osmia bicornis. |
| Wild animals | Our study did not involve wild animals. |
| Field-collected samples | We stored pollen and bee samples at -20C before screening for pesticide compounds included in the Swedish national monitoring scheme, following established protocols at the analytical laboratory. |
| Ethics oversight | No ethical approval is required for insect collections in Sweden. |

Note that full information on the approval of the study protocol must also be provided in the manuscript.

## Human research participants

Policy information about studies involving human research participants

| Population characteristics | *Describe the covariate-relevant population characteristics of the human research participants (e.g. age, gender, genotypic information, past and current diagnosis and treatment categories). If you filled out the behavioural & social sciences study design questions and have nothing to add here, write "See above."* |
| Recruitment | *Describe how participants were recruited. Outline any potential self-selection bias or other biases that may be present and how these are likely to impact results.* |
| Ethics oversight | *Identify the organization(s) that approved the study protocol.* |

Note that full information on the approval of the study protocol must also be provided in the manuscript.

## Clinical data

Policy information about clinical studies

All manuscripts should comply with the ICMJE guidelines for publication of clinical research and a completed CONSORT checklist must be included with all submissions.

| Clinical trial registration | *Provide the trial registration number from ClinicalTrials.gov or an equivalent agency.* |
| Study protocol | *Note where the full trial protocol can be accessed OR if not available, explain why.* |
| Data collection | *Describe the settings and locales of data collection, noting the time periods of recruitment and data collection.* |
| Outcomes | *Describe how you pre-defined primary and secondary outcome measures and how you assessed these measures.* |

## Dual use research of concern

Policy information about dual use research of concern

### Hazards

Could the accidental, deliberate or reckless misuse of agents or technologies generated in the work, or the application of information presented in the manuscript, pose a threat to:

| No | Yes | |
|----|-----|---|
| ☐ | ☐ | Public health |
| ☐ | ☐ | National security |
| ☐ | ☐ | Crops and/or livestock |
| ☐ | ☐ | Ecosystems |
| ☐ | ☐ | Any other significant area |

## Experiments of concern

Does the work involve any of these experiments of concern:

| No | Yes | |
|----|-----|--|
| ☐ | ☐ | Demonstrate how to render a vaccine ineffective |
| ☐ | ☐ | Confer resistance to therapeutically useful antibiotics or antiviral agents |
| ☐ | ☐ | Enhance the virulence of a pathogen or render a nonpathogen virulent |
| ☐ | ☐ | Increase transmissibility of a pathogen |
| ☐ | ☐ | Alter the host range of a pathogen |
| ☐ | ☐ | Enable evasion of diagnostic/detection modalities |
| ☐ | ☐ | Enable the weaponization of a biological agent or toxin |
| ☐ | ☐ | Any other potentially harmful combination of experiments and agents |

# ChIP-seq

## Data deposition

☐ Confirm that both raw and final processed data have been deposited in a public database such as GEO.

☐ Confirm that you have deposited or provided access to graph files (e.g. BED files) for the called peaks.

**Data access links**
*May remain private before publication.*

*For "Initial submission" or "Revised version" documents, provide reviewer access links. For your "Final submission" document, provide a link to the deposited data.*

**Files in database submission**

*Provide a list of all files available in the database submission.*

**Genome browser session**
(e.g. UCSC)

*Provide a link to an anonymized genome browser session for "Initial submission" and "Revised version" documents only, to enable peer review. Write "no longer applicable" for "Final submission" documents.*

## Methodology

**Replicates**

*Describe the experimental replicates, specifying number, type and replicate agreement.*

**Sequencing depth**

*Describe the sequencing depth for each experiment, providing the total number of reads, uniquely mapped reads, length of reads and whether they were paired- or single-end.*

**Antibodies**

*Describe the antibodies used for the ChIP-seq experiments; as applicable, provide supplier name, catalog number, clone name, and lot number.*

**Peak calling parameters**

*Specify the command line program and parameters used for read mapping and peak calling, including the ChIP, control and index files used.*

**Data quality**

*Describe the methods used to ensure data quality in full detail, including how many peaks are at FDR 5% and above 5-fold enrichment.*

**Software**

*Describe the software used to collect and analyze the ChIP-seq data. For custom code that has been deposited into a community repository, provide accession details.*

# Flow Cytometry

## Plots

Confirm that:

☐ The axis labels state the marker and fluorochrome used (e.g. CD4-FITC).

☐ The axis scales are clearly visible. Include numbers along axes only for bottom left plot of group (a 'group' is an analysis of identical markers).

☐ All plots are contour plots with outliers or pseudocolor plots.

☐ A numerical value for number of cells or percentage (with statistics) is provided.

## Methodology

**Sample preparation**

*Describe the sample preparation, detailing the biological source of the cells and any tissue processing steps used.*

**Instrument**

*Identify the instrument used for data collection, specifying make and model number.*

| Software | *Describe the software used to collect and analyze the flow cytometry data. For custom code that has been deposited into a community repository, provide accession details.* |
|---|---|
| Cell population abundance | *Describe the abundance of the relevant cell populations within post-sort fractions, providing details on the purity of the samples and how it was determined.* |
| Gating strategy | *Describe the gating strategy used for all relevant experiments, specifying the preliminary FSC/SSC gates of the starting cell population, indicating where boundaries between "positive" and "negative" staining cell populations are defined.* |

☐ Tick this box to confirm that a figure exemplifying the gating strategy is provided in the Supplementary Information.

# Magnetic resonance imaging

## Experimental design

| Design type | *Indicate task or resting state; event-related or block design.* |
|---|---|
| Design specifications | *Specify the number of blocks, trials or experimental units per session and/or subject, and specify the length of each trial or block (if trials are blocked) and interval between trials.* |
| Behavioral performance measures | *State number and/or type of variables recorded (e.g. correct button press, response time) and what statistics were used to establish that the subjects were performing the task as expected (e.g. mean, range, and/or standard deviation across subjects).* |

## Acquisition

| Imaging type(s) | *Specify: functional, structural, diffusion, perfusion.* |
|---|---|
| Field strength | *Specify in Tesla* |
| Sequence & imaging parameters | *Specify the pulse sequence type (gradient echo, spin echo, etc.), imaging type (EPI, spiral, etc.), field of view, matrix size, slice thickness, orientation and TE/TR/flip angle.* |
| Area of acquisition | *State whether a whole brain scan was used OR define the area of acquisition, describing how the region was determined.* |

Diffusion MRI     ☐ Used     ☐ Not used

## Preprocessing

| Preprocessing software | *Provide detail on software version and revision number and on specific parameters (model/functions, brain extraction, segmentation, smoothing kernel size, etc.).* |
|---|---|
| Normalization | *If data were normalized/standardized, describe the approach(es): specify linear or non-linear and define image types used for transformation OR indicate that data were not normalized and explain rationale for lack of normalization.* |
| Normalization template | *Describe the template used for normalization/transformation, specifying subject space or group standardized space (e.g. original Talairach, MNI305, ICBM152) OR indicate that the data were not normalized.* |
| Noise and artifact removal | *Describe your procedure(s) for artifact and structured noise removal, specifying motion parameters, tissue signals and physiological signals (heart rate, respiration).* |
| Volume censoring | *Define your software and/or method and criteria for volume censoring, and state the extent of such censoring.* |

## Statistical modeling & inference

| Model type and settings | *Specify type (mass univariate, multivariate, RSA, predictive, etc.) and describe essential details of the model at the first and second levels (e.g. fixed, random or mixed effects; drift or auto-correlation).* |
|---|---|
| Effect(s) tested | *Define precise effect in terms of the task or stimulus conditions instead of psychological concepts and indicate whether ANOVA or factorial designs were used.* |

Specify type of analysis:     ☐ Whole brain     ☐ ROI-based     ☐ Both

| Statistic type for inference<br>(See Eklund et al. 2016) | *Specify voxel-wise or cluster-wise and report all relevant parameters for cluster-wise methods.* |
|---|---|
| Correction | *Describe the type of correction and how it is obtained for multiple comparisons (e.g. FWE, FDR, permutation or Monte Carlo).* |

## Models & analysis

| n/a | Involved in the study |
|-----|-----------------------|
| ☐ | ☐ Functional and/or effective connectivity |
| ☐ | ☐ Graph analysis |
| ☐ | ☐ Multivariate modeling or predictive analysis |

**Functional and/or effective connectivity**

*Report the measures of dependence used and the model details (e.g. Pearson correlation, partial correlation, mutual information).*

**Graph analysis**

*Report the dependent variable and connectivity measure, specifying weighted graph or binarized graph, subject- or group-level, and the global and/or node summaries used (e.g. clustering coefficient, efficiency, etc.).*

**Multivariate modeling and predictive analysis**

*Specify independent variables, features extraction and dimension reduction, model, training and evaluation metrics.*

