## [Peer Review File · Nature Ecology & Evolution]

Peer Review Information

Journal: Nature Ecology & Evolution

Manuscript Title: Ecological traits interact with landscape context to determine bees' pesticide risk

Corresponding author name(s): Jessica Knapp

Editorial Notes:

Reviewer Comments & Decisions:

Decision Letter, initial version:

6th June 2022

Dear Dr Knapp,

Your manuscript entitled "Ecological traits interact with landscape context to determine bees' pesticide risk" has now been seen by 3 reviewers, whose comments are attached. The reviewers have raised a number of concerns which will need to be addressed before we can offer publication in Nature Ecology & Evolution. We will therefore need to see your responses to the criticisms raised and to some editorial concerns, along with a revised manuscript, before we can reach a final decision regarding publication.

We therefore invite you to revise your manuscript taking into account all reviewer and editor comments. Please highlight all changes in the manuscript text file [OPTIONAL: in Microsoft Word format].

* If you have not done so already please begin to revise your manuscript so that it conforms to our Article format instructions at <http://www.nature.com/natecolevol/info/final-submission>. Refer also to any guidelines provided in this letter.

2[REDACTED]

Nature Ecology & Evolution is committed to improving transparency in authorship. As part of our efforts in this direction, we are now requesting that all authors identified as 'corresponding author' on published papers create and link their Open Researcher and Contributor Identifier (ORCID) with their account on the Manuscript Tracking System (MTS), prior to acceptance. ORCID helps the scientific community achieve unambiguous attribution of all scholarly contributions. You can create and link your ORCID from the home page of the MTS by clicking on 'Modify my Springer Nature account'. For more information please visit www.springernature.com/orcid.

[REDACTED]

Reviewer expertise:

Reviewer #1: Pollinators, pesticides

Reviewer #2: Pollinators, pesticides

Reviewer #3: Pollinators, pesticides

Reviewers' comments:

Reviewer #1 (Remarks to the Author):

I think this is a very interesting study, with clear significance for the management of pollinator species and relative differences in risk of agrochemical exposure related to bee specific traits. I have a number of things I want clarified but I think this is a potentially important paper which will have a high impact in the area Ben Woodcock.

2Abstract

L33: By highest risk I assume you are referring to pesticides here. If so is this from a strict ecotoxicology perspective, i.e a risk quotient or something like that / or is this in a more general idea of risk.

L32-34: Also I would split this sentence in two, one dealing with pesticide sensitivity, the other with land sue, the way its structured they don't really combine well.

L34L:L same point with risk here.

L36: This sounds good, but I am not really following what you mean. By identify are you referring to the type or class of agrochemical compound, re are you referring to the bee species (the start of the sentence is talking about bee specific traits). This needs a bit of work for clarity.

Introduction and discussion are well written and crafted. My issues relate more to clarification of the methods.

Methods

L86 delete 'our'

L86: What is site separation?

L87: I have circled back to this, but I am a bit ignorant of the clover system you have. Is this for forage or seed - Coming from the uk this would not receive agrochemicals. You might need to explain this one more.

L90-92: So are sites chosen along a gradient of agricultural land use - if so what is the range of this gradient for each crop.

L93: Farmers managed crops conventionally, with the exception of one field of each focal crop which was managed organically.

L93: Might need to expand on what they are typically spraying here - see comment about clover, but generally true. If its simple enough put in main text, otherwise summarise and put in suppl. Material.

L97: where these of the same variety - i.e. similar genetic stock of Apis. Also how big were the colonies, nucs, 1 year old, established. Was this consistent.

L100: explain why not at the clover sites - I am assuming its because they cant forage on clover.

L102-109: So how many sampled were taken and over what time periods. It's a bit unclear. Is it just a single 24h period. How did you decide on this - I.e. was this timed to coincide with known peaks of agrochemical usage on the crop?

L110 - ok , covered here. However, in terms of the timing of after pesticide applications, which applications, herbivces, insecticides, fungicides? Some more information is needed here. Did this vary between sites.

L115-117: So how many chemicals - can you give some idea, i.e. how many fungicides, insecticides, herbicides. Does this cover known types that will have been applied to these crops..

L119-129: Sounds good - its unclear how many samples this was done for - when you say part do you mean part of each pollen sample taken? I am guessing you do, but maybe make this clear. It could be interpreted as a subset of all sampled collected.

L131: Maybe justify the basis of those scales by giving some examples of typical flight foraging ranges of Apis, solitary and Bombus spp.

L139: I am happy with that approach

L144: Good. Its amore ecotoxicology based approach. However, is this a strict toxic unit as you are LD50 is based dose rate per bee, and as such for Risk quotients which this seems a derivation of you are trying to work out consumption rate of an life stage relevant time period. That is to say I though a TU was the concentration (e.g. dose rate per bee) of a given pesticide required to cause a given

toxicological endpoint (LD50 which is in a.i. / bee). I don't think you need to change anything as you are looking at relative differences and the adjustment makes sense. While I think you could convert to daily consumption rates for honeybees I don't think you could do this effectively of Apis or Osmia so this makes sense. I would just check exactly what a TU unit is and clarify this in response to that. N.b. its also worth noting at some point that the LD50 that you are using to scale these is based on honeybee responses, and that these are not necessarily what you would find for Osmia and Bombus which likely have difference toxicokinetic and dynamic responses to agrochemical exposure. That said there is nothing you can do here, as Apis is still largely the model system.

L152: Ok you define what you mean as risk - good. Like I said you might need to tighten up some definitions, but the approach seems good.

L155: good

L158_162. I like this approach. I think it may be worth mentioning somewhere that this is assuming concentration addition of products with similar mode of action. There are other approaches for compounds with different modes of action. Also maybe also mention the to the potential for unexpected synergistic effects (e.g.azole fungicides and insecticides).

SO I just looked at your statement in the abstract 'We provide foraging trait and landscape-dependent information on the identity, level and frequency of pesticide exposure needed to meet policy goals to reduce pesticide risk and ambitions for more realistic risk assessments'. TO what extent is the latter part of this strictly true from an ecotox perspective - i.e. I am not convinced you are strictly speaking providing a toxic unit as its not scaled in terms of consumption which is the unit that LD50 is based on. Have a look at Guidance for Assessing Pesticide Risks to Bees by US EPA, and in particular their Bee-REX model. Again I think you just need to be careful, and I just want you to be aware of how ecotoxicologists have very hard views on what risk strictly is and how these different metrics are interpreted. Just make certain the inferences you make from a policy perspective are robust , or at least qualified.

Reviewer #2 (Remarks to the Author):

Here, the authors investigate the impact of crop type and landscape context on the exposure of three bee species to pesticide residues (and its associated risk) using a creative and novel design, representing a substantial amount of work. The paper helps to bridge the gap in understanding between impacts of pesticides in a lab setting, and exposure in the field. As such it is a very valuable addition to the literature.

4nature portfolio

A fantastic paper with an unbelievable amount of work, effort and thought put into it. A pleasure to read a short paper with such interesting results and conclusions, and one that handles a complex subject very well.

General:

The paper is framed around the concept of functional traits, but ultimately there is only one species/replicate per functional trait group. Therefore it is not clear whether differences were driven by the functional traits in question or other inter-species differences, and to truly examine differences in functional trait types either a number of species per category or a gradient would be needed. Realistically, this paper is more just a comparison between three key bee species important for crop pollination. This is still an important and interesting question and it may be a simpler message within the paper in general to introduce it this way rather than add the complication of functional traits.

Abstract: Clear and concise, very good.

33: Previous order of foraging range went 'extensive, intermediate, limited', now here it is 'extensive, limited, intermediate'. Given limited had the same response as intermediate, maybe just swap their order around for the readers sake.

36: 'Extrapolates precautiously' is a confusing wording, maybe try rephrasing it (same throughout).

Introduction: A solid introduction, well written and sets up the paper nicely.

40: *semi-natural habitat area

48: Use of biodiversity terminology for pesticide diversity could be confusing. Maybe switch to 'amount and diversity of pesticides...' to differentiate the language more clearly.

51: "habitats that usually provide refuge from pesticides will decrease", this isn't actually a conclusion of your prior sentence as the word 'thus' implies. The below rewrite makes it a logically sequential point while retaining your meaning. "Thus, as agriculture intensifies the remaining areas of semi-natural habitats that usually provide refuge from pesticides will increasingly becoming a potential source of exposure".

58: Worth specifying that it's pesticide exposure each time, as it will help a reader follow.

Figure 1a: Would not some bumblebee researchers take slight offence to the term 'weakly social' to describe bumblebees? Up to the authors to use the terms they want to, but some researchers would argue they are firmly social.

Figure 1b: I don't understand what visual clarity the baseline adds. Not saying it's wrong, I just don't understand it. Please explain why it helps the reader understand you graph.

nature portfolio

Figure 1c and d: Surely the centre of the circle should be where the colonies should be, both the extensive and limited foragers. That way the circles represent their ranges? I can see the intention behind these figures and very much do like the visualisation. But I don't think they convey the message you're intending. Can I suggest that you iterate on them, showing them to naïve people, and see if they can understand and extract your central message from them. Visually they're great, clean cut and simple.

68: 'moderate' is a directional word, implying a downward reduction. Your testing allows it to go either direction, so 'control', 'affect' or 'influence' are all directionally neutral words which would be better choices.

68: Very long sentence, maybe break it up.

Figure 2a and b: Lovely figures.

Figure 2c: Making this one larger would be helpful, I had to squint to read it. Same with the legend.

Figure 2d: 'Proportion of pollen collected' would be more obvious as an axis title. Again, larger text please, or just larger graph.

Methods: Good methods, covers a lot of ground and has done a good job deciding what information to retain and which to exclude. The methods are not wholly replicable based on the description presented here, but this is not the authors fault, merely a factor of the word count for such a huge amount of work. I've tried to be sparing in recommending more information be added, as a fine balance has clearly been struck here.

In general, the chemical analysis of pollen and nectar to detect pesticide residues needs to be described in more detail (either in main manuscript or supplementary). For example, at the moment it is not clear what compounds were looked for, and what the LODs were. Knowledge of both of these is crucial to allow proper interpretation of the findings.

94: Which was managed organically.

103: Please add more details on how the pollen was taken from the returning bumblebees.

L103-105: How was this collected in relation to crop spray? Samples collected directly after spray are likely quite different to those even a few days later. See comment on L108-109 also in relation to this.

L108: How does the number of pollen loads for bumblebees here relate to the number of samples, i.e. what is a sample? Also, as written, pollen seems to have been sampled from 24 sites, 6 bumblebee colonies per site, ~20 foragers per colony...this adds up to almost double the 1309 that was actually sampled? Please clarify here how many samples were actually taken and what the relationship between pollen load and "sample" is?

L108-109: 16 samples of *Osmia* pollen is very low across 24 sites, and represents significantly less sampling than for the other two species. This is a limitation which should be more explicit in the

6discussion.

110: The use of pollen and nectar is fantastic, and over three species and three crops is very impressive.

L110-114: It is not clear whether this sampling was part of the sampling described in the first paragraph or in addition? Clarify this. Also perhaps I missed this, but it's not clear where the results of this section are presented? Make this clearer

L124: "Pollen was counted in 7-20 rows" – why the large variation? Surely this resulted in quite different sampling effort between sites which would likely influence total counts?

L126: Why is the *Malus domestica* pollen description labelled "Prunus" (a different genus) when all other groupings are labelled the relevant family or genus name?

L135: It is common knowledge that pesticides are used in urban areas and, in particular as this is a less regulated type of use, use could be quite high in some situations. The amount of pesticide usage in the landscape here is assumed to be related to the proportion of agricultural land only. This at least needs to be acknowledged somewhere.

135-137: Too technical for a reader outside the field to understand, please rewrite more plainly.

L140-142: Ranges would be useful to present here instead of, or in addition to, means as this would give the reader more of an idea of the gradient.

145: What about substances that don't have LD50's. Glyphosate and most herbicides don't have LD50s, they are written as LD50>200µg a.i. per bee. How were these handled?

153: Where did you get your LD50s from?

153: Given only nectar and pollen are quantified, and both represent oral exposure, it's hard to see why the contact and oral values were averaged, rather than just using the oral value. Personally, I really can't see any justification for this, as the two exposure routes are so distinct, and only oral really related to the residue work conducted. Please substantively explain to the reader why you chose to do this or use oral only. Also, would not a LC50 be more apt, the residues being returned are measured as concentrations, yet you're using acute values. Toxicologically it's a big assumption to equate acute and oral. LC50s are also widely available for *Apis*, again please justify or change.

L154-157: I understand why the authors use LD50s for *Apis mellifera* (as there is a shortage of data on other groups) but it's a pity the data aren't available for the others as extrapolating across species is probably not realistic.

L172: Strictly, LMMs do not make the assumption of normality and homogeneity of variance. What is important is how residuals look, and this is usually what decisions on transformation could/should be made on. Consider rewording to take this into account.

7Results: Generally good, but I think lacking in detail. Again, more so a function of the journal requirements than the authors choices though. Other papers have done just the residue data work (3 species and 3 crops) and had that be the entirety of their paper, yet it is given relatively little space here. I would like to have seen more detail given, perhaps that should come in the supplementary materials though, ideally also the whole dataset. I'm not seeing a statement of data accessibility. For a dataset this size, with so much nuance lost in the concise format, accessible data is essential. I think the authors have done a good job in understanding what is interesting about their work and is worthy of presenting.

209: Surprised not to see a list of these somewhere, surely that's interesting data to present.

General: The naming of the supplementary files is confusing, and I struggled to jump back and forth between them. Please consider how best to lay them out and title them from a readers perspective. The supplementary files are relied upon very heavily in a paper like this, so this is crucial.

213: Formally $\mu\text{g}/\text{kg}$ is more accurate, but it's up to the authors to decide on their unit choice.

214: This is a personal preference of mine, so action it however you will, but I find the use of numerous acronyms complicates reading a paper. So, when I got to 'PRQ' it took a while to remember. Maybe just spell things out more for the reader.

218: "the three-way interaction", 'a' implies there's more than one available, and if that's true you need to specify which one.

229: see above.

254: 'Risk was higher in pollen than nectar'- see larger commentary on this in the discussion.

Discussion: Very well written discussion, not overly focussing on any one area, nicely summarises the results. There are some areas of nuance that need to be explored here though, and this is mainly the source of my comments below.

265: Here, in the discussion, is where the nuance of pollen vs nectar should be delved into properly. As it stands it's reading like pollen is the big risk for bees, but that's not reflective of the consumption of each substance.

Risk = hazard * exposure. The hazard estimate is as good as can be, based on existing data for the numerous substances detected. The exposure component though is based only on concentration. Yet, it is more accurate to model exposure as Exposure = concentration * consumption. Adults eat little pollen, but lots of nectar, hence their exposure is drastically different despite the concentrations.

Of course, larvae eat lots of pollen, but the model is built and framed around adult bees. It's also untrue to say larval pollen consumption is 1:1 with adult nectar.

It would be possible to redo essentially the entire modelling work, redefining exposure by using published values on bees consumption of either nectar and pollen. That's very much outside the scope of a reviewer comment though, and instead I think there should be a frank discussion reflecting that the risk calculation does not serve the pollen vs nectar comparison well. This should be included in the discussion, as well as referenced in the results (for readers who don't make it to the discussion). An alternative is to just remove the pollen vs nectar risk comparison, as it's a flawed comparisons anyhow and not a central conclusion. This would free up space for other results to be presented. It is of course up to the authors to decide how to action these comments, or even disagree with my interpretation.

288: Throughout there's the assumption of agriculture intensifying meaning pesticide use is going up. But pesticide use globally plateaued in 2018 (FAO REF), and in some nations like the UK it is in decline (REF). Complicating matters is the various units of pesticide use. In Sweden it may be going up (I don't know), but the paper isn't framed around Sweden as it's framed very broadly. Readers will just accept your statement of increasing pesticide use, but this isn't conveying the nuanced reality, e.g. the EU has committed to a 50% reduction in pesticide use in the next decade or so.

Similarly, there's the statement that increasingly semi-natural habitats are becoming scarce. Please substantiate this with evidence. Many would look at CAP, and the Farm to Fork strategy, as well as global analogous measures and think maybe we'll actually increase semi-natural areas over the coming few decades. (see <https://ourworldindata.org/peak-agriculture-land>)

Please address these two points substantively, as it's rather a central point of your paper. This means not just in the discussion, but also in the introduction (in less detail).

298: A quick reminder on what a focal crop is?

312: Was imidacloprid legal for outdoor use at the time of sampling? If not expand on this, if so, consider putting a '(since banned)' to indicate to the reader it was legal at the time.

318: If imidacloprid was banned in 2018 and you sampled in 2019, expand on this. People love talking about neonicotinoids and a finding that they persist after a ban would be a key point people will be left to ponder.

320: Sulfoxaflor has just joined the neonicotinoids in being banned in the EU, so remove the references relating to it (58) as they are now misleading.

General: Using LD50s isn't a perfect measure of hazard. We know that LD50s aren't particularly good at actually predicting hazard because they only look at mortality, not sublethal effects. Given the entire risk term is built on mortality specific data, I would like to see an acknowledgement that mortality is not fitness, and fitness is the ultimate metric.

<https://doi.org/10.1038/s41559-020-1194-6> is a good perspective on this. Beyond this, a one

sentence acknowledgement that the toxicity of herbicides, fungicides, co-formulants and adjuvants is underrepresented by LD50s because of LD50s mortality focus (respective to a more comprehensive assessment of hazard), could help pull the literature away from the focus on just insecticides.

322: Pollen vs nectar comparison.

329: ERA is an acronym most won't remember.

348: Your final two sentences are very true, and something I wholly agree with them, but they're not a central outcome of the paper. This paper has a goldmine of data and conclusions, I'd suggest focussing on one of them to end, versus focussing on ERA. Wholly at the authors discretion of course.

Figure 3: The shades of yellow and brown aren't visually distinct.

Figure 4: Red and green are not very colour-blind friendly colours. Try using an online tool to view the figure from a deuteranopia perspective. Could be applied to all colour schemes throughout really, as they're not that distinct colours.

References: Check for consistency as there are some issues (e.g. doi's given for some and not others, some titles have all words capitalised and some not, species names should be in italics etc)

Reviewer #3 (Remarks to the Author):

Manuscript background:

The authors present a novel and interesting approach to address the role of bee life history traits, specifically foraging range, in predicting bee pollinator exposure to insecticides in three agricultural crop settings and landscape contexts where pesticides are used, with the goal of understanding if these factors alone or in combination influence pesticide risk for the purposes of informing ecological risk assessment.

Major comments:

Very interesting, exciting and meticulous work! This work is a really valuable contribution to the discipline.

Minor comments:

1. The authors might consider referencing the following paper, which supports statements in this manuscript of the contributions of pesticide drift to contamination of non-cultivated crop pollen and risks this exposure presents to pollinators:

Long, E. Y., & Krupke, C. H. (2016). Non-cultivated plants present a season-long route of pesticide exposure for honey bees. *Nature communications*, 7(1), 1-12.

2. Supplementary Methods, Lines 1-14: There are a few technical pieces of information missing from this section that would increase the transparency and credibility of the pesticide residue quantification process: a) What were the limits of detection for focal compounds quantified with LC and GC,

10respectively? b) What specific analytical instruments were used (typically model names and numbers are reported)? c) Were pollen samples from each species pooled for residue analysis? d) Were technical replicates of pesticide residue samples tested? Reporting this information would strengthen the manuscript.

3. Line 582, 585 and 593: Spelling out MCR in these figure captions would improve interpretation and help figures stand alone.

*****END*****

Author Rebuttal to Initial comments

Ecological traits interact with landscape context to determine bees' pesticide risk

Response to reviewer comments

We want to thank the reviewers for their thoughtful comments, which have greatly improved the quality of the manuscript. We have substantially revised the manuscript to provide greater clarity, especially regarding our ecological approach to bees' environmental pesticide risk. We respond below, and line numbers refer to those in the clean version of the document.

Reviewer #1 (Remarks to the Author):

I think this is a very interesting study, with clear significance for the management of pollinator species and relative differences in risk of agrochemical exposure related to bee specific traits. I have a number of things I want clarified but I think this is a potentially important paper which will have a high impact in the area Ben Woodcock.

Response: Thank you, we are pleased you like the manuscript and have worked to improve clarity.

L33: By highest risk I assume you are referring to pesticides here. If so is this from a strict ecotoxicology perspective, i.e a risk quotient or something like that / or is this in a more general idea of riskl.

Response: You raise an important point we now address throughout the manuscript. We take an ecological, general perspective on bees' environmental pesticide-related risk, defined by additive, toxicity-weighted exposure. This metric allows us to explore how bees' activity patterns, determined by their trait compositions, affect the level and identity of compounds encountered and thus disentangle bees' ecology from toxicology. We have used a similar metric as in Rundlöf et al. (2022), where we also confirmed a reduced bee reproduction as the environmental pesticide-related risk increased – indicating the relevance and applicability of this metric.

L32-34: Also I would split this sentence in two, one dealing with pesticide sensitivity, the other with land sue, the way its structured they don't really combine well.

Response: We have divided this sentence into two "We found that extensive foragers experienced the highest pesticide-related risk. However, only limited and intermediate foragers responded to landscape context – experiencing lower risk with less agricultural land." – L32-35.

L34L:L same point with risk here.

Response: We now say 'pesticide risk.'

L36: This sounds good, but I am not really following what you mean. By identify are you referring to the type or class of agrochemical compound, re are you referring to the bee species (the start of the sentence is talking about bee specific traits). This needs a bit of work for clarity.

Response: We have clarified this sentence -" We provide foraging trait and landscape-dependent information on the identity, level and frequency of pesticide compounds that bees encounter - steps toward more holistic and realistic risk assessment and essential information for meeting policy goals to reduce pesticide risk" – L36-39.

Introduction and discussion are well written and crafted. My issues relate more to clarification of the methods.

Response: Thank you, and we have specified how we have clarified the methodology in the proceeding points.

L86 delete 'our'

Response: done.

L86: What is site separation?

Response: We have added this information. "All sites were more than 6km apart, except two clover sites 2km apart." – L99.

L87: I have circled back to this, but I am a bit ignorant of the clover system you have. Is this for forage or seed - Coming from the uk this would not receive agrochemicals. You might need to explain this one more.

Response: We now clarify "red clover grown for seed production" and have additionally provided information on the national recommendations of pesticide use for pest control in the three focal crops. "The national pest management recommendations for 2019 included 26 active ingredients in oilseed, 32 in apple and 14 in clover seed in acaricide (2 active ingredients), fungicide (20), herbicide (20), and insecticide (13) products (see Table S1 for details)." – L92-95.

L90-92: So are sites chosen along a gradient of agricultural land use - if so what is the range of this gradient for each crop.

Response: Yes, they are, and we now provide this range in text. "The average (\pm SD) proportion of agricultural land was $74\% \pm 24$ (range: 29 - 95%) for oilseed, $52\% \pm 29$ (6 - 85%) for apples, and $66\% \pm 20$ (44 - 93%) for clover. All sites were more than 6km apart, except for two clover sites, 2km apart." – L97-99.

L93: Farmers managed crops conventionally, with the exception of one field of each focal crop which was managed organically.

Response: Changed as suggested.

L93: Might need to expand on what they are typically spraying here - see comment about clover, but generally true. If its simple enough put in main text, otherwise summarise and put in supl. Material.

Response: We now provide information on the national pesticide use recommendations for the three focal crops "The national pest management recommendations for 2019 included 26 active ingredients in oilseed, 32 in apple and 14 in clover seed in acaricide (2 active ingredients), fungicide (20), herbicide (20), and insecticide (13) products (see Table S1 for details)." – L92-95.

L97: where these of the same variety - i.e. similar genetic stock of Apis. Also how big were the colonies, nucs, 1 year old, established. Was this consistent.

Response: We now provide further detail for the sentinels of three bee species in the supplementary information referenced on L109. "The *Apis mellifera* colonies were prepared at the end of April 2019 in local Swedish colony size (Lågnormal; inside dimensions, 382 x 382 x 230 mm; about ¾ the size of a full-frame Langstroth hive) with two frames of brood, two frames of nectar and pollen stores, four frames of drawn comb and two frames of foundation; about 0,5 kg bees and a laying, open-mated 1-2-year-old queen of mixed genetic stock (primarily *A.m. carnica* with traces of *A.m. ligustica* and *A.m. mellifera*). The colonies were treated for varroa with two strips of Apistan (tau-fluvalinate) between 1 September-13 October 2018 and a single treatment of 3.2% oxalic acid in sugar syrup in November 2018. Varroa management or treatment was not applied during the 2019 experiments, although varroa development was monitored. The colonies were free from American foulbrood (AFB), European foulbrood (EFB) and tracheal mites (*Acarapis woodi*), the three primary reportable diseases in Sweden. The colonies were supplied with extra space as required and managed to prevent swarming. None of the colonies swarmed during the experiments, although one colony did lose its queen without replacement. Standard colonies of *Bombus terrestris* were sourced from Biobest Biological systems (Belgium). Each colony contained a queen and about 80 worker bees plus brood. We removed the sugar water provision to make the bees forage for nectar and pollen, i.e. most resembling wild bumblebees. Cocoons of *Osmia bicornis* were sourced from Wildbiene & Partner (Switzerland) and stored hibernating at 4°C before a diapause break at 10°C. The cocoons were then placed in an emergence tube within the nesting unit for release. The nesting units were designed by Red BeeHive (UK) and consisted of three plastic trap nests, filled with a

central emergence tube surrounded by cardboard nesting tubes, mounted on a wooden pole at 1-1.5 m high off the ground."

L100: explain why not at the clover sites – I am assuming its because they cant forage on clover.

Response: We now clarify, "We did not place *O. bicornis* in red clover fields as their phenologies do not overlap (Fig. 2c)." – L109-110.

L102-109: So how many sampled were taken and over what time periods. It's a bit unclear. Is it just a single 24h period. How did you decide on this – I.e. was this timed to coincide with known peaks of agrochemical usage on the crop?

Response: We now explain that sampling intervals coincide with crop bloom, "We sampled pollen from (1) *A. mellifera* using pollen traps attached to two hives for 24 hours, (2) *B. terrestris* by capturing foragers (~20 across all six colonies) and euthanising them on dry ice as they returned to their colonies, and (3) multiple *O. bicornis* brood cells collected by females over the second half of the bloom period. We sampled pollen from *A. mellifera* and *B. terrestris* at two sampling intervals, coinciding with (1) the peak of crop bloom and (2) after crop bloom and for *O. bicornis* only at the end of crop bloom (evenly from all the available pollen). In total, we collected 48 samples (595 g) of *A. mellifera*-, 44 samples (11 g) of *B. terrestris*-, and 16 samples (70 g) of *O. bicornis*-collected pollen. During and after bloom samples were pooled for both *A. mellifera* and *B. terrestris*, resulting in 24 samples of *A. mellifera*-, 22 samples of *B. terrestris*- (all colonies died at two sites), and 16 samples of *O. bicornis*-collected pollen. We did not pool *O. bicornis* pollen for the bloom period since this species already combined pollen provisions on our behalf." – L112-123.

L110 - ok , covered here. However, in terms of the timing of after pesticide applications, which applications, herbivces, insecticides, fungicides? Some more information is needed here. Did this vary between sites.

Response: We now provide an additional table (Table S2) outlining the pesticide applications that form the basis for the spatiotemporally matched pollen and nectar samples.

L115-117: So how many chemicals – can you give some idea, i.e. how many fungicides, insecticides, herbicides. Does this cover known types that will have been applied to these crops..

15Response: We now state that 120 compounds were screened for and include a list of all screened compounds (Table S3) and note if the Swedish Board of Agriculture recommended their use in our focal crops in 2019 (Table S1). All pesticide compounds known to be applied at our study sites in 2019 (Table S2) were screened, except for Dithianon (Delan WG), which was applied to one apple site.

L119-129: Sounds good - its unclear how many samples this was done for - when you say part do you mean part of each pollen sample taken? I am guessing you do, but maybe make this clear. It could be interpreted as a subset of all sampled collected.

Response: Yes, we now clarify that it was part of each pollen sample (L135)

L131: Maybe justify the basis of those scales by giving some examples of typical flight foraging ranges of Apis, solitary and Bombus spp.

Response: We now cite bees' average foraging capacities, listed in Figure 1. "We analysed the landscape surrounding our sites at multiple spatial scales (1000, 1500, and 2000 m, corresponding to bees' average foraging capacities (Figure 1))". – L148.

L139: I am happy with that approach

Response: Thank you.

L144: Good. Its amore ecotoxicology based approach. However, is this a strict toxic unit as you are LD50 is based dose rate per bee, and as such for Risk quotients which this seems a derivation of you are trying to work out consumption rate of an life stage relevant time period. That is to say I though a TU was the concentration (e.g. dose rate per bee) of a given pesticide required to cause a given toxicological endpoint (LD50 which is in a.i. / bee). I don't think you need to change anything as you are looking at relative differences and the adjustment makes sense. While I think you could convert to daily consumption rates for honeybees I don't think you could doo this effectively of Apis or Osmia so this makes sense. I would just check exactly what a TU unit is and clarify this in response to that. N.b. its also worth noting at some point that the LD50 that you are using to scale these is based on honeybee responses, and that these are not necessarily what you would find for Osmia and Bombus which likely have difference toxicokinetic and dynamic responses to

agrochemical exposure. That said there is nothing you can do here, as Apis is still largely the model system.

Response: We agree and have changed 'toxic-unit' to 'toxicity-weighted exposure' to describe our ecological approach to calculating bees' pesticide-related risk in the environment. Using the same LD50 values across bee species, we disentangle the bees' ecology from toxicology to explore relative differences in bees' activity patterns in intersection with pesticide use.

L152: Ok you define what you mean as risk - good. Like I said you might need to tighten up some definitions, but the approach seems good.

Response: Thank you.

L155: good

L158_162. I like this approach. I think it may be worth mentioning somewhere that this is assuming concentration addition of products with similar mode of action. There are other approaches for compounds with different modes of action. Also maybe also mention the to the potential for unexpected synergistic effects (e.g. azole fungicides and insecticides).

Response: Thank you, and we have now added a clause stating that synergistic effects may occur – "Then, following a concentration addition approach - the recommended default for mixture environmental risk assessment (even though some compound classes may synergise⁴⁰)⁴¹", "- L162-164.

SO I just looked at your statement in the abstract 'We provide foraging trait and landscape-dependent information on the identity, level and frequency of pesticide exposure needed to meet policy goals to reduce pesticide risk and ambitions for more realistic risk assessments'. TO what extent is the latter part of this strictly true from an ecotox perspective - i.e. I am not convinced you are strictly speaking providing a toxic unit as its not scaled in terms of consumption which is the unit that LD50 is based on. Have a look at Guidance for Assessing Pesticide Risks to Bees by US EPA, and in particular their Bee-REX model. Again I think you just need to be careful, and I just want you to be aware of how ecotoxicologists have very hard views on what risk strictly is and how these different metrics are

interpreted. Just make certain the inferences you make from a policy perspective are robust , or at least qualified.

Response: Thank you for pointing this out - you raise a good point in relation to toxic units. We now clarify that we use an additive measure of toxicity-scaled exposure and thus not strictly a 'toxic unit'. Furthermore, we have toned down our last sentence to reflect that our work is a step toward policy and risk assessment goals. " We provide foraging trait and landscape-dependent information on the identity, level and frequency of pesticides that bees encounter - steps toward more holistic and realistic risk assessment and essential information for meeting policy goals to reduce pesticide risk."– L36-39.

Reviewer #2:

Here, the authors investigate the impact of crop type and landscape context on the exposure of three bee species to pesticide residues (and its associated risk) using a creative and novel design, representing a substantial amount of work. The paper helps to bridge the gap in understanding between impacts of pesticides in a lab setting, and exposure in the field. As such it is a very valuable addition to the literature.

A fantastic paper with an unbelievable amount of work, effort and thought put into it. A pleasure to read a short paper with such interesting results and conclusions, and one that handles a complex subject very well.

Response: Thank you for your nice feedback; we are very pleased you recognised the quality of the work and enjoyed reading the manuscript.

General:

The paper is framed around the concept of functional traits, but ultimately there is only one species/replicate per functional trait group. Therefore it is not clear whether differences were driven by the functional traits in question or other inter-species differences, and to truly examine differences in functional trait types either a number of species per category or a gradient would be needed. Realistically, this paper is more just a comparison between three key bee species important for crop pollination. This is still an important and interesting question and it may be a simpler message within the paper in general to introduce it this way rather than add the complication of functional traits.

Response: Whilst we agree that our manuscript would benefit from adding more bee species, including functional traits enabled us to provide a framework for predicting other bee species' exposure and risk, useful for scientists, risk assessors and policy-makers.

Abstract: Clear and concise, very good.

Response: Thank you.

33: Previous order of foraging range went 'extensive, intermediate, limited', now here it is 'extensive, limited, intermediate'. Given limited had the same response as intermediate, maybe just swap their order around for the readers sake.

Response: Changed as suggested.

36: 'Extrapolates precautiously' is a confusing wording, maybe try rephrasing it (same throughout).

Response: We agree and have changed this to " Pesticide risk correlated among bee species and between food sources and was greatest in *A. mellifera*-collected pollen" – L35-36

Introduction: A solid introduction, well written and sets up the paper nicely.

Response: Thank you.

40: *semi-natural habitat area

Response: Changed as suggested.

48: Use of biodiversity terminology for pesticide diversity could be confusing. Maybe switch to 'amount and diversity of pesticides...' to differentiate the language more clearly.

Response: Changed as suggested.

51: "habitats that usually provide refuge from pesticides will decrease", this isn't actually a conclusion of your prior sentence as the word 'thus' implies. The below rewrite makes it a logically sequential point while retaining your meaning. "Thus, as agriculture intensifies the remaining areas of semi-natural habitats that usually provide refuge from pesticides will increasingly becoming a potential source of exposure".

Response: Thank you; we have changed it to your suggestion.

58: Worth specifying that it's pesticide exposure each time, as it will help a reader follow.

Response: Changed as suggested.

Figure 1a: Would not some bumblebee researchers take slight offence to the term 'weakly social' to describe bumblebees? Up to the authors to use the terms they want to, but some researchers would argue they are firmly social.

Response: Whilst we agree that bumblebees are social, our terminology reflects bumblebees' intermediate sociality between honeybees and solitary bees, in contrast to solitary bees not forming colonies and lacking honeybees' advanced communication of rewarding resources.

Figure 1b: I don't understand what visual clarity the baseline adds. Not saying it's wrong, I just don't understand it. Please explain why it helps the reader understand you graph.

Response: The baseline simply shows that pesticide exposure and risk will increase with agricultural intensification, proportional to the area of agricultural land within the bees' foraging range (L613-615).

Figure 1c and d: Surely the centre of the circle should be where the colonies should be, both the extensive and limited foragers. That way the circles represent their ranges? I can see the intention behind these figures and very much do like the visualisation. But I don't think they convey the message you're intending. Can I suggest that you iterate on them, showing them to naïve people, and see if they can understand and extract your central message from them. Visually they're great, clean cut and simple.

Response: We have added further detail to the legend of figure 1 to explain that the grey triangles and squares are the locations of foraging bees rather than nests, with the central nests now depicted with a "X". "Figure 1. A trait-based, spatially explicit framework for bees' pesticide exposure and risk. We describe (a) three sets of bees' foraging traits (based on^{9,18,69 70}): 'extensive', 'intermediate' and 'limited', in relation to (b) landscape context, as demonstrated in (c) low-intensity and (d) high-intensity landscapes, whereby extensive (grey square) and limited (grey triangle) foragers move between habitat types within their respective foraging ranges (concentric circles relative to X – the central nests). Our baseline assumption (b, black circles) is that pesticide exposure and risk will increase with agricultural intensification, proportional to the area of agricultural land within the bees' foraging range (c and d, concentric circles). We expect bees with the largest foraging range, 'extensive' foragers, to receive the highest pesticide exposure and risk independent of landscape context (b, line intercept; c and d, grey squares). However, as agriculture intensifies, the proportion of agricultural land within the bees foraging range increases, and the likelihood of foraging on contaminated food increases. Therefore, we expect

20'limited' foragers to be disproportionately more at risk from pesticide exposure as agricultural land expands (b, line slope; c and d, grey triangles)." – L608-621.

68: 'moderate' is a directional word, implying a downward reduction. Your testing allows it to go either direction, so 'control', 'affect' or 'influence' are all directionally neutral words which would be better choices.

Response: Changed to "alter" here and in the abstract.

68: Very long sentence, maybe break it up.

Response: We agree the sentence is long and have now reformulated it slightly to reduce the repeated mentioning of cropping system and landscape both in the middle and end of the sentence. "To test whether foraging traits affect bees' exposure and risk in different landscape contexts, we assayed pesticide residues in pollen and nectar collected by *A. mellifera*, *B. terrestris* and *O. bicornis*, representing extensive, intermediate and limited foragers, across three sequentially blooming crops (Fig. 1-2)." – L69-72.

Figure 2a and b: Lovely figures.

Response: Thank you.

Figure 2c: Making this one larger would be helpful, I had to squint to read it. Same with the legend.

Response: Changed as suggested.

Figure 2d: 'Proportion of pollen collected' would be more obvious as an axis title. Again, larger text please, or just larger graph.

Response: Changed as suggested.

Methods: Good methods, covers a lot of ground and has done a good job deciding what information to retain and which to exclude. The methods are not wholly replicable based on the description presented here, but this is not the authors fault, merely a factor of the word count for such a huge amount of work. I've tried to be sparing in recommending more information be added, as a fine balance has clearly

been struck here.

Response: Thank you for helping us strike this balance.

In general, the chemical analysis of pollen and nectar to detect pesticide residues needs to be described in more detail (either in main manuscript or supplementary). For example, at the moment it is not clear what compounds were looked for, and what the LODs were. Knowledge of both of these is crucial to allow proper interpretation of the findings.

Response: We now include a list of all screened compounds and their detection limits (Table S3) as well as further detail to the Supplementary Methods on our analytical approach.

94: Which was managed organically.

Response: Changed.

103: Please add more details on how the pollen was taken from the returning bumblebees.

Response: We have added further detail to specify that the *B. terrestris* foragers were captured and euthanised on dry ice as they returned to their colonies. "*B. terrestris* by capturing foragers (~20 across all colonies) and euthanising them on dry ice as they returned to their colonies" – L113-114.

L103-105: How was this collected in relation to crop spray? Samples collected directly after spray are likely quite different to those even a few days later. See comment on L108-109 also in relation to this.

Response: We collected samples in relation to crop bloom, which we now clarify. " We sampled pollen from *A. mellifera* and *B. terrestris* at two sampling intervals, coinciding with (1) the peak of crop bloom and (2) after crop bloom and for *O. bicornis* only, at the end of crop bloom (evenly from all the available pollen). In total, we collected 48 samples (387 g) of *A. mellifera*-, 44 samples (15 g) of *B. terrestris*-, and 16 samples (70 g) of *O. bicornis*- collected pollen. During and after bloom samples were pooled for both *A. mellifera* and *B. terrestris*, resulting in 24 samples of *A. mellifera*-, 22 samples of *B. terrestris*- (all colonies died at two sites), and 16 samples of *O. bicornis*-collected pollen. We did not pool *O. bicornis* pollen for the bloom period since this species combined pollen provisions on our behalf." – L112-123.

L108: How does the number of pollen loads for bumblebees here relate to the number of samples, i.e. what is a sample? Also, as written, pollen seems to have been sampled from 24 sites, 6 bumblebee

colonies per site, ~20 foragers per colony...this adds up to almost double the 1309 that was actually sampled? Please clarify here how many samples were actually taken and what the relationship between pollen load and "sample" is?

Response: We have clarified that the 20 *B. terrestris* foragers were collected across all colonies. There were two sampling rounds (during and after crop bloom) from 22 sites (because the bumblebee colonies died at two sites, we had 24 in total). Thus, we have 44 samples, each containing pollen loads from around 20 bees. We have edited the total weight of pollen accordingly, as the last number was incorrect.— " We sampled pollen from (1) *A. mellifera* using pollen traps attached to two hives for 24 hours, (2) *B. terrestris* by capturing foragers (~20 across all six colonies) and euthanising them on dry ice as they returned to their colonies, and (3) multiple *O. bicornis* brood cells collected by females over the second half of the bloom period. We sampled pollen from *A. mellifera* and *B. terrestris* at two sampling intervals, coinciding with (1) the peak of crop bloom and (2) after crop bloom and for *O. bicornis* only at the end of crop bloom (evenly from all the available pollen). In total, we collected 48 samples (595 g) of *A. mellifera*-, 44 samples (11 g) of *B. terrestris*-, and 16 samples (70 g) of *O. bicornis*- collected pollen. During and after bloom samples were pooled for both *A. mellifera* and *B. terrestris*, resulting in 24 samples of *A. mellifera*-, 22 samples of *B. terrestris*- (all colonies died at two sites), and 16 samples of *O. bicornis*-collected pollen. We did not pool *O. bicornis* pollen since this species already combined pollen provisions for the bloom period on our behalf.- L112-123.

L108-109: 16 samples of *Osmia* pollen is very low across 24 sites, and represents significantly less sampling than for the other two species. This is a limitation which should be more explicit in the discussion.

Response: We acknowledge that *O. bicornis* was sampled less, since the species was not placed at the clover sites because of non-overlapping phenologies. Since clover was the less risky focal crop, we probably provide an overestimation of the risk of pesticides for this species relative to the others – something that we bring up in the discussion even if it there is related to the use of sentinels vs natural occurring populations of the species (L323-327). We have reworded this section to clarify the number of samples between bee species – "During and after bloom samples were pooled for both *A. mellifera* and *B. terrestris*, resulting in 24 samples of *A. mellifera*-, 22 samples of *B. terrestris*- (all colonies died at two sites), and 16 samples of *O. bicornis*-collected pollen. We did not pool *O. bicornis* pollen for the bloom period since this species already combined pollen provisions for the on our behalf" – L119-123.

110: The use of pollen and nectar is fantastic, and over three species and three crops is very impressive.

Response: Thank you.

L110-114: It is not clear whether this sampling was part of the sampling described in the first paragraph or in addition? Clarify this. Also perhaps I missed this, but it's not clear where the results of this section are presented? Make this clearer

Response: We have clarified that we used (additional) paired samples to explore the risk between pollen and nectar. –" To compare residues between nectar and pollen, we sampled additional returning foragers of *A. mellifera* (n = ~100 individuals per sample) and *B. terrestris* (n = ~20 individuals per sample) 1-2, 4-6 and 12-16 days after a known pesticide application at four oilseed, two apple and seven clover sites (Table S2). Corbicular pollen and nectar stomach content were collected from these foragers to produce paired pollen and nectar samples for each site and collection time point (n = 54)." – L124-129.

L124: "Pollen was counted in 7-20 rows" – why the large variation? Surely this resulted in quite different sampling effort between sites which would likely influence total counts?

Response: We now clarify our methodology to show that we aimed to count >400 pollen grains and thus needed to count a variable number of rows depending on the density of pollen in the gel and thereafter calculated a proportional pollen use. "We identified (using a pollen reference library at the Department of Biology (Lund) and Sawyer³³) and counted >400 pollen grains per slide (7-20 rows, 163 µm wide across the slide) using 400x magnification." – L139-141.

L126: Why is the *Malus domestica* pollen description labelled "Prunus" (a different genus) when all other groupings are labelled the relevant family or genus name?

Response: We have now labelled all groups to the relevant genus name and provided further information in Table S4.

L135: It is common knowledge that pesticides are used in urban areas and, in particular as this is a less regulated type of use, use could be quite high in some situations. The amount of pesticide usage in the landscape here is assumed to be related to the proportion of agricultural land only. This at least needs to be acknowledged somewhere.

nature portfolio

Response: We agree that pesticide use in urban areas could be an important driver of bees' pesticide risk; however, our focus was on agricultural pesticide use, which we now reiterate in the methodology. "We classified land cover categories into two groups: agricultural land (all types of agricultural use, such as annual crops, orchards, leys and semi-natural grasslands) and non-agricultural land (including forest, urban areas and water bodies) since our focus was on bees' pesticide exposure and risk from agricultural pesticide use. Indeed, bees' pesticide exposure is higher in rural compared to urban areas³⁸." – L149-154.

135-137: Too technical for a reader outside the field to understand, please rewrite more plainly.

Response: Changed to – "We also calculated the proportion of the focal crop in the radii and the average field size."- L154.

L140-142: Ranges would be useful to present here instead of, or in addition to, means as this would give the reader more of an idea of the gradient.

Response: Changed to add ranges in addition.

145: What about substances that don't have LD50's. Glyphosate and most herbicides don't have LD50s, they are written as LD50>200µg a.i. per bee. How were these handled?

Response: We now provide this information- "LD₅₀s were rounded down to their value when expressed as 'greater than'³⁵." – L177-178.

153: Where did you get your LD50s from?

Response: LD50 values are from the Pesticide Properties Data Base, University of Hertfordshire, as cited in the text (L162).

153: Given only nectar and pollen are quantified, and both represent oral exposure, it's hard to see why the contact and oral values were averaged, rather than just using the oral value. Personally, I really can't see any justification for this, as the two exposure routes are so distinct, and only oral really related to the residue work conducted. Please substantively explain to the reader why you chose to do this or use oral only. Also, would not a LC50 be more apt, the residues being returned are measured as

concentrations, yet you're using acute values. Toxicologically it's a big assumption to equate acute and oral. LC50s are also widely available for *Apis*, again please justify or change.

Response: We average oral and contact LD50s to provide an overall indicator of the mixture's toxicity and have clarified our rationale - " We averaged each compound's acute oral and contact LD₅₀³⁷ to provide an overall indicator of toxicity, reflective of how bees encounter pesticides in the landscape, i.e. moving contaminated food in contact with their bodies for oral consumption⁴⁰." – L170-172.

Unfortunately, we could not find extensive *Apis* LC50s and thus used LD50s from the Pesticide Properties Data Base, consistent with other researchers, e.g. Sanchez-Bayo and Goka (2014).

L154-157: I understand why the authors use LD50s for *Apis mellifera* (as there is a shortage of data on other groups) but it's a pity the data aren't available for the others as extrapolating across species is probably not realistic.

Response: We agree that it is a shame that more toxicity data is not available. However, dividing by a common LD50 allowed us to explore relative differences in risk between bee species.

L172: Strictly, LMMs do not make the assumption of normality and homogeneity of variance. What is important is how residuals look, and this is usually what decisions on transformation could/should be made on. Consider rewording to take this into account.

Response: Changed.

Results: Generally good, but I think lacking in detail. Again, more so a function of the journal requirements than the authors choices though. Other papers have done just the residue data work (3 species and 3 crops) and had that be the entirety of their paper, yet it is given relatively little space here. I would like to have seen more detail given, perhaps that should come in the supplementary materials though, ideally also the whole dataset. I'm not seeing a statement of data accessibility. For a dataset this size, with so much nuance lost in the concise format, accessible data is essential. I think the authors have done a good job in understanding what is interesting about their work and is worthy of presenting.

Response: Thank you. Our data will be made freely available on Figshare when this manuscript is published, and we now include a data availability statement. In the meantime, the data can be viewed via a private link: <https://figshare.com/s/ee4270fb7b463fc6a5a6>

209: Surprised not to see a list of these somewhere, surely that's interesting data to present.

Response: We provide a list in Table S1 and have included citations. " Across bee species (*A. mellifera*, *B. terrestris*, and *O. bicornis*) and crops (oilseed, apple, and clover) for both food sources (pollen and nectar), a total of 53 compounds were detected, including 24 fungicides, 19 herbicides, five insecticides, two acaricides, two metabolites of herbicides and one metabolite of a fungicide. We detected more compounds in pollen samples from oilseed sites (42, n = 40) than apple (36, n = 36) and clover sites (25, n = 32). The four compounds with the greatest compound-specific risk were insecticides (Table 1), but some herbicides and fungicides also ranked highly (Table S5). Herbicides and fungicides comprise 80% of total detections and 65% of total residues (in $\mu\text{g}/\text{kg}$), yet unsurprisingly insecticides represented the majority of pesticide risk, accounting for over 99% of the compound-specific risk.— L234-243.

General: The naming of the supplementary files is confusing, and I struggled to jump back and forth between them. Please consider how best to lay them out and title them from a readers perspective. The supplementary files are relied upon very heavily in a paper like this, so this is crucial.

Response: We agree this was confusing and have worked to restructure the supplementary more logically, including a contents page with hyperlinks that meets the journal's guidelines.

213: Formally $\mu\text{g}/\text{kg}$ is more accurate, but it's up to the authors to decide on their unit choice.

Response: Thank you for bringing this to our attention. We have changed throughout.

214: This is a personal preference of mine, so action it however you will, but I find the use of numerous acronyms complicates reading a paper. So, when I got to 'PRQ' it took a while to remember. Maybe just spell things out more for the reader.

Response: Changed throughout.

218: *"the three-way interaction", 'a' implies there's more than one available, and if that's true you need to specify which one.

Response: Changed

229: see above.

Response: Changed

254: 'Risk was higher in pollen than nectar'- see larger commentary on this in the discussion.

Response: We now clarify that risk refers to environmental pesticide-related risk, i.e. bees' exposure (scaled by toxicity) due to their activity intersecting pesticide use and thus not related to consumption. "Environmental pesticide risk was higher in pollen than in nectar.." – L282.

Discussion: Very well written discussion, not overly focussing on any one area, nicely summarises the results. There are some areas of nuance that need to be explored here though, and this is mainly the source of my comments below.

Response: Thank you, we have added further detail to the discussion to address your proceeding points.

265: Here, in the discussion, is where the nuance of pollen vs nectar should be delved into properly. As it stands it's reading like pollen is the big risk for bees, but that's not reflective of the consumption of each substance.

Response: We agree, and further to our previous response, better clarify our ecological approach and its associated caveats to estimating pesticide risk in the discussion. " However, pesticide exposure and our ecological indicator of pesticide risk do not account for species-specific processes past the pesticide use-bee activity intersection, such as consumption within the nest or indirect effects that could affect bees' fitness – important considerations when moving from exposure to effect in environmental risk assessment⁷². L373-377.

Risk = hazard * exposure. The hazard estimate is as good as can be, based on existing data for the numerous substances detected. The exposure component though is based only on concentration. Yet, it is more accurate to model exposure as Exposure = concentration * consumption. Adults eat little pollen, but lots of nectar, hence their exposure is drastically different despite the concentrations. Of course, larvae eat lots of pollen, but the model is built and framed around adult bees. It's also untrue to say larval pollen consumption is 1:1 with adult nectar.

Response: Whilst including consumption data would provide another estimate of exposure and risk, data is scarce for *Bombus terrestris* and *Osmia bicornis*, and we did not want to assume values and thus add uncertainty. Furthermore, our focus was on environmental exposure, i.e. initial exposure outside of

and as it enters a nest, which we now clarify throughout the manuscript.

It would be possible to redo essentially the entire modelling work, redefining exposure by using published values on bees consumption of either nectar and pollen. That's very much outside the scope of a reviewer comment though, and instead I think there should be a frank discussion reflecting that the risk calculation does not serve the pollen vs nectar comparison well. This should be included in the discussion, as well as referenced in the results (for readers who don't make it to the discussion). An alternative is to just remove the pollen vs nectar risk comparison, as it's a flawed comparisons anyhow and not a central conclusion. This would free up space for other results to be presented. It is of course up to the authors to decide how to action these comments, or even disagree with my interpretation.

Response: We include the pollen and nectar comparison because our terminology reflects bees' environmental pesticide-related risk, i.e. risk outside of the nest and thus does not consider consumption, now clarified throughout the manuscript. Quantifying the identity, level and frequency of pesticide compounds that bees encounter in this way is a step toward policy goals to reduce pesticide risk and ambitions for a more realistic risk assessment.

288: Throughout there's the assumption of agriculture intensifying meaning pesticide use is going up. But pesticide use globally plateaued in 2018 (FAO REF), and in some nations like the UK it is in decline (REF). Complicating matters is the various units of pesticide use. In Sweden it may be going up (I don't know), but the paper isn't framed around Sweden as it's framed very broadly. Readers will just accept your statement of increasing pesticide use, but this isn't conveying the nuanced reality, e.g. the EU has committed to a 50% reduction in pesticide use in the next decade or so.

Response: We only discuss intensive agriculture as a practice that includes, among other things, high reliance on pesticides and reduced semi-natural habitats (Foley et al. 2005) and have checked and changed our wording throughout. For example, "However, limited foragers may become disproportionately more exposed in intensive agricultural landscapes, where there is an increased likelihood of contamination in diminished semi-natural habitats (Fig. 1b; line slope)." – L66-68.

Similarly, there's the statement that increasingly semi-natural habitats are becoming scarce. Please substantiate this with evidence. Many would look at CAP, and the Farm to Fork strategy, as well as global analogous measures and think maybe we'll actually increase semi-natural areas over the coming few decades. (see <https://ourworldindata.org/peak-agriculture-land>) Please address these two points

substantively, as it's rather a central point of your paper. This means not just in the discussion, but also in the introduction (in less detail).

Response: Similarly, we only discuss reduced semi-natural habitats as a process of agricultural intensification. We do not attempt to discuss trends in agricultural practices over time as this is beyond the scope of our study. We now start the introduction by defining agricultural intensification as "Agricultural intensification includes concomitant reductions in semi-natural areas and increased reliance on pesticides^{1,2}" (L41-42).

298: A quick reminder on what a focal crop is?

Response: Added.

312: Was imidacloprid legal for outdoor use at the time of sampling? If not expand on this, if so, consider putting a '(since banned)' to indicate to the reader it was legal at the time.

Response: Yes, the EU had banned imidacloprid for outdoor use at the time of sampling, which we address in the proceeding point.

318: If imidacloprid was banned in 2018 and you sampled in 2019, expand on this. People love talking about neonicotinoids and a finding that they persist after a ban would be a key point people will be left to ponder.

Response: We now bring this to our readers' attention " In the EU, chemical restrictions (imidacloprid 2018, thiacloprid 2020 and indoxacarb 2022) are regulatory moves in this direction⁶²⁻⁶⁴, even if residues persist (like imidacloprid in our study⁶⁵) or new compounds with similar risk profiles enter the market in the future⁶⁵⁻⁶⁷." –L359-362.

320: Sulfoxaflor has just joined the neonicotinoids in being banned in the EU, so remove the references relating to it (58) as they are now misleading.

Response: Removed.

General: Using LD50s isn't a perfect measure of hazard. We know that LD50s aren't particularly good at actually predicting hazard because they only look at mortality, not sublethal effects. Given the entire risk term is built on mortality specific data, I would like to see an acknowledgement that mortality is not

fitness, and fitness is the ultimate metric. <https://doi.org/10.1038/s41559-020-1194-6> is a good perspective on this. Beyond this, a one sentence acknowledgement that the toxicity of herbicides, fungicides, co-formulants and adjuvants is underrepresented by LD50s because of LD50s mortality focus (relative to a more comprehensive assessment of hazard), could help pull the literature away from the focus on just insecticides.

Response: We use LD50s as an information-rich, general indicator of toxicity to scale pesticide exposure (similar to Rundlöf et al. 2022, indicating a negative correlation between this risk index and bee reproduction). Furthermore, pesticide effects, including fitness, are beyond the scope of our study, although we now expand on this in the discussion. " However, pesticide exposure and our ecological indicator of pesticide risk do not account for species-specific processes past the pesticide use-bee activity intersection, such as consumption within the nest or indirect effects that could affect bees' fitness – important considerations when moving from exposure to effect in environmental risk assessment⁷². – L373-377.

322: Pollen vs nectar comparison.

Response: We now clarify that we mean environmental risk.

329: ERA is an acronym most won't remember.

Response: Changed throughout.

348: Your final two sentences are very true, and something I wholly agree with them, but they're not a central outcome of the paper. This paper has a goldmine of data and conclusions, I'd suggest focussing on one of them to end, versus focussing on ERA. Wholly at the authors discretion of course.

Response: We are pleased that you can see the many possible conclusions of this work. We chose to end on ERA because pesticide policy is critical to pesticide use and thus bees' pesticide exposure and risk.

Figure 3: The shades of yellow and brown aren't visually distinct.

Response: We asked independent colleagues to review these colours and concluded that they were visually distinct. Furthermore, we checked the COBLIS colour blindness simulator, and they are colour-blind friendly.

Figure 4: Red and green are not very colour-blind friendly colours. Try using an online tool to view the figure from a deuteranopia perspective. Could be applied to all colour schemes throughout really, as they're not that distinct colours.

Response: We checked our figures in the COBLIS colour blindness simulator and agree they are not colour-blind friendly. We found that decreasing the opacity improved our figures' dichromatic views (e.g., red/green blindness); see examples below. We also limited the use of greens in Fig 2. Both adjustments improved the colour-blind friendliness of images while keeping the colour scheme consistent between our conceptual figure (Fig. 1) and results figures (Figs. 2-S9).

References: Check for consistency as there are some issues (e.g. doi's given for some and not others, some titles have all words capitalised and some not, species names should be in italics etc)

Response: Checked and changed.

Reviewer #3

Manuscript background:

The authors present a novel and interesting approach to address the role of bee life history traits, specifically foraging range, in predicting bee pollinator exposure to insecticides in three agricultural crop settings and landscape contexts where pesticides are used, with the goal of understanding if these factors alone or in combination influence pesticide risk for the purposes of informing ecological risk assessment.

Response: Thank you, we are pleased you see the novelty and application of this work.

Major comments:

Very interesting, exciting and meticulous work! This work is a really valuable contribution to the discipline.

Response: Thank you.

Minor comments:

1. The authors might consider referencing the following paper, which supports statements in this manuscript of the contributions of pesticide drift to contamination of non-cultivated crop pollen and risks this exposure presents to pollinators:

Long, E. Y., & Krupke, C. H. (2016). Non-cultivated plants present a season-long route of pesticide exposure for honey bees. *Nature communications*, 7(1), 1-12.

Response: Added.

2. Supplementary Methods, Lines 1-14: There are a few technical pieces of information missing from this section that would increase the transparency and credibility of the pesticide residue quantification process: a) What were the limits of detection for focal compounds quantified with LC and GC, respectively? b) What specific analytical instruments were used (typically model names and numbers are

reported)? c) Were pollen samples from each species pooled for residue analysis? d) Were technical replicates of pesticide residue samples tested? Reporting this information would strengthen the manuscript.

Response: We have added further detail about the chemical analysis to the supplementary information, including a list of all screened compounds with their detection limits.

3. Line 582, 585 and 593: Spelling out MCR in these figure captions would improve interpretation and help figures stand alone.

Response: Changed.

Decision Letter, first revision:

9th September 2022

Dear Dr Knapp,

Your manuscript entitled "Ecological traits interact with landscape context to determine bees' pesticide risk" has now been seen again by 2 of the previous reviewers, whose comments are attached.

As you can see, Referee #1 is satisfied with the changes, but Referee #2 still has some minor comments that need addressing -- in particular they feel that two problematic sentences still remain in the text discussing pollen vs nectar, that they feel do not fully address their concerns. As requested by this Referee, we suggest removing these sentences (Lines 282-285 and 340 -342) from the text to avoid any potential confusion.

We will also need to see your responses to the other minor criticisms raised, along with a revised manuscript, before we can reach a final decision regarding publication.

We therefore invite you to revise your manuscript taking into account all reviewer and editor comments. Please highlight all changes in the manuscript text file in Microsoft Word format.

We are committed to providing a fair and constructive peer-review process. Do not hesitate to contact us if there are specific requests from the reviewers that you believe are technically impossible or

34unlikely to yield a meaningful outcome.

* If you have not done so already please begin to revise your manuscript so that it conforms to our Article format instructions at <http://www.nature.com/natecolevol/info/final-submission>. Refer also to any guidelines provided in this letter.

[REDACTED]

Nature Ecology & Evolution is committed to improving transparency in authorship. As part of our efforts in this direction, we are now requesting that all authors identified as 'corresponding author' on published papers create and link their Open Researcher and Contributor Identifier (ORCID) with their account on the Manuscript Tracking System (MTS), prior to acceptance. ORCID helps the scientific community achieve unambiguous attribution of all scholarly contributions. You can create and link your ORCID from the home page of the MTS by clicking on 'Modify my Springer Nature account'. For more information please visit www.springernature.com/orcid.

[REDACTED]Reviewers' comments:

Reviewer #1 (Remarks to the Author):

Thank you for your excellent revision and thorough response to my comments on the original version. I am happy with what you have done to address the issues that I originally raised and now feel that I can recommend this manuscript for publications. I think this will make an excellent contribution to this journal. Ben Woodcock

Reviewer #2 (Remarks to the Author):

The response to the reviewer comments has been thorough and well thought through. Lots of changes have been made, clearly in response to diligent consideration of the reviewer comments. There are a couple of more substantive points that have not been addressed satisfactorily, and we have elaborated on them in this review. For any points from the original review we still wanted to consider at this stage, we have given the original reviewer comment, followed by author response and then our second round comment below. In addition we have a generic comment on the pollen vs nectar question, and two minor comments at the end.

SPECIFIC RESPONSES:

Original reviewer comment:

145: What about substances that don't have LD50's. Glyphosate and most herbicides don't have LD50s, they are written as LD50>200µg a.i. per bee. How were these handled?

Response:

We now provide this information- "LD50s were rounded down to their value when expressed as 'greater than'35." – L177-178.

Second round reviewer comment:

The value of 200µg isn't an estimate, it's a wholly arbitrary value. We just stop testing at 200µg because it becomes hard to work with such concentrated substances. You need to state that this is a directional assumption i.e., the model will systematically overestimate the risk from these substances. The rounding down is fine, but given it essentially is just assigning an arbitrary value to the substance, this needs a mention that it'll overestimate risk.

Original reviewer comment:

153: Given only nectar and pollen are quantified, and both represent oral exposure, it's hard to see why the contact and oral values were averaged, rather than just using the oral value. Personally, we really can't see any justification for this, as the two exposure routes are so distinct, and only oral really related to the residue work conducted. Please substantively explain to the reader why you chose to do this or use oral only. Also, would not a LC50 be more apt, the residues being returned are measured as concentrations, yet you're using acute values. Toxicologically it's a big assumption to equate acute and oral. LC50s are also widely available for Apis, again please justify or change.

Response:

36We average oral and contact LD50s to provide an overall indicator of the mixture's toxicity and have clarified our rationale - " We averaged each compound's acute oral and contact LD5037 to provide an overall indicator of toxicity, reflective of how bees encounter pesticides in the landscape, i.e. moving contaminated food in contact with their bodies for oral consumption40." – L170-172.

Unfortunately, we could not find extensive Apis LC50s and thus used LD50s from the Pesticide Properties Data Base, consistent with other researchers, e.g. Sanchez-Bayo and Goka (2014).

Second round reviewer comment:

We're not convinced by this as a justification. Bees contact exposure doesn't really happen when they move nectar/pollen back to the nest. It mostly happens when foraging and they are passing through a spray cloud or contact spray residues on surfaces. It also happens in a much much more diffuse manner when touching stuff like soil or pollen, but this is not ever going to cause acute toxicity analogous to a contact LD50. Simply, pollen isn't a source of contact exposure that translates to contact LD50s. Perhaps this wording "We averaged each compound's acute oral and contact LD5037 to provide an overall indicator of toxicity, reflective of how bees encounter pesticides in the landscape and the multiple exposure routes they can be exposed through" would more accurately surmise this. Removing the pollen example will actually strengthen the argument.

Original reviewer comment:

General: The naming of the supplementary files is confusing, and I struggled to jump back and forth between them. Please consider how best to lay them out and title them from a readers perspective. The supplementary files are relied upon very heavily in a paper like this, so this is crucial.

Response:

We agree this was confusing and have worked to restructure the supplementary more logically, including a contents page with hyperlinks that meets the journal's guidelines.

Second round reviewer comment:

Maybe it's still the journal's system, but all the files came through with numerical names and we had to open them one by one and rename them something sensible. Just have another look at the actual file names and if it's not the journal's fault give them more logical names. Maybe it is an error on our side, and apologies if so.

Pollen vs nectar.

Generic Second round reviewer comment addressing the Pollen vs Nectar comparison:

This is the main area we are not happy with the edits. It doesn't seem like the text has changed to reflect the points we made. Perhaps we were not clear on what we were explaining. It's only 2 sentences in the manuscript, so it's not a pervasive issue. We would advocate for just removing the two sentences entirely to avoid this issue stalling the (otherwise fabulous) paper's publication. Lines 282-285 and 340 -342 are the sentences of interest here.

We asked for a frank discussion of how nectar and pollen contribute to bee toxicity differently and how the modelling cannot answer this question. What has been changed amounts to some minor edits which unfortunately use dense scientific language to qualify the statements.

Put simply, we are not convinced that the pollen vs nectar claim is true, and think it is outside the scope of the model the answer this question. The attempts to qualify the claims have not been satisfactory, nor has the longer explanation in the reviewer response been sufficient to allay our concerns.

The paper claims "Environmental pesticide risk was higher in pollen than in nectar" and "we found that exposure and risk were higher in pollen than in nectar". These two claims, in our eyes, are false (or at least heavily misleading to a reader).

Firstly, the edit to change 'risk' to 'Environmental pesticide risk' is not effective in detailing what the intended meaning. The use of technical language obfuscates the point and puts the emphasis on the reader remembering the terminology and distinction, rather than spelling it out for them. If a reader does not know what the distinction between 'Environmental pesticide risk' and 'risk' is, they will assume they are the same, making the sentence untrue. Readers are seeing the paper for the first time, so it is important to assume they know nothing about it, may not even read whole sections, and thus act accordingly.

Second. To state that exposure to pesticides is higher in pollen than nectar is not true. What was found was higher concentrations in pollen. Then the assumption of equal consumption was made. With this assumption, your statements are true, but the assumption is false. The statement "we found that exposure [was] higher in pollen" is untrue. It is possible to contest this by arguing bees face equivalent exposure to pollen and nectar. If sufficient references and data can be presented to convince us this is the case, this will suffice. But given our understanding of exposure to pesticides, we find this unlikely to happen.

The model used is fantastic at answering the central questions of the paper. However, it's not suitable for a comparison of pollen and nectar. It would need additional terms and to include the differential consumption (or an inclusive exposure term that accounts for non-consumption exposure i.e., contact toxicity).

As noted in the prior review, we do not believe it reasonable to ask for the model to be changed. We do ask that the inferences from the model be changed to fit only what is reasonable to infer from it. The pollen vs nectar 'environmental pesticide risk' comparison is not reasonable to infer.

In an already very data rich and expansive paper, we see no real reason to include these datapoints. They are a methodologically flawed comparisons. Simply cut them and spend those words elaborating on the strong and rigorous conclusions. It would be possible to elaborate on and properly contextualise the results, but doing that (as we have below) raises the question of why they are interesting to include. Modifications to the text as below may help clarify this issue:

"Our model found environmental pesticide risk was higher in pollen than in nectar (Fig. 5a; $T = -10.55$, $df = 93.86$, $P < 0.01$), although this assumes equal exposure and consumption of both matrices. This assumption is unlikely to be true given bees consume very different quantities of pollen and nectar, and their ecotoxicology is very different."

"...we found that exposure and risk were higher in pollen than in nectar, although this may be a factor of the model not accounting for differential consumption/exposure to the two mediums, which are not equivalent."

Apologies for labouring this point.

Author Response:

We include the pollen and nectar comparison because our terminology reflects bees' environmental pesticide-related risk, i.e. risk outside of the nest and thus does not consider consumption, now clarified throughout the manuscript. Quantifying the identity, level and frequency of pesticide compounds that bees encounter in this way is a step toward policy goals to reduce pesticide risk and ambitions for a more realistic risk assessment.

Second round reviewer comment:

This does not make much sense to us. How is pollen causing risk outside of the nest? It's not ingested (outside the nest) in pollen basket species, and it will cause near zero contact toxicity (it cannot penetrate the cuticle). If the paper is discussing risk outside the nest, pollen will contribute essentially no risk whatsoever. This interpretation of what the model is measuring is odd and not conveyed well in the paper.

How can risk not consider consumption? Risk = hazard (accurately quantified in your model) x Exposure. Exposure = concentration x (consumption for oral exposure OR contact amount for contact exposure). You neither quantify consumption nor contact amount. Hence risk cannot be quantified. What is then being calculated is simply hazard * concentration, which is a parameter which will align with risk, but will not allow for a medium-to-medium comparison (as medium-to-medium the consumption/contact amount changes). To call it risk is an acceptable misnomer so long as long as there is no medium-to-medium comparison which highlights the weakness of the comparison.

It feels as if there is a misunderstanding of what 'risk' means in a ecotoxicology sense, or that the redefinition of it in the paper is causing this confusion.

Original reviewer comment:

288: Throughout there's the assumption of agriculture intensifying meaning pesticide use is going up. But pesticide use globally plateaued in 2018 (<https://www.fao.org/3/cb1329en/CB1329EN.pdf>), and in some nations like the UK it is in decline. Complicating matters is the various units of pesticide use. In Sweden it may be going up (I don't know), but the paper isn't framed around Sweden as it's framed very broadly. Readers will just accept your statement of increasing pesticide use, but this isn't conveying the nuanced reality, e.g. the EU has committed to a 50% reduction in pesticide use in the next decade or so.

Response:

We only discuss intensive agriculture as a practice that includes, among other things, high reliance on pesticides and reduced semi-natural habitats (Foley et al. 2005) and have checked and changed our wording throughout. For example, "However, limited foragers may become disproportionately more exposed in intensive agricultural landscapes, where there is an increased likelihood of contamination

39nature portfolio

in diminished semi-natural habitats (Fig. 1b; line slope)." – L66-68.

Second round reviewer comment:

The argument that agriculture is intensifying is still an argument made in your piece. Wordings like Line 52- "Thus, as agriculture intensifies, the remaining areas of semi-natural habitats that usually provide refuge from pesticides may increasingly become a potential source of exposure" explicitly state agriculture is intensifying.

If this wording is to be retained, you need to justify this with data. However, FAO data (<https://www.fao.org/3/cb1329en/CB1329EN.pdf>) shows global pesticide use has plateaued and land area for agricultural use has declined. So we would recommend revisiting these statements; for example reference could be made just to agricultural intensification currently rather than in the future.

Minor comments:

L44 – a word is missing

Figure 2d: what is displayed on the X axis is not clear – consider adding in axis explanation

*****END*****

Author Rebuttal, first revision:

Reviewer #1 (Remarks to the Author):

Thank you for your excellent revision and thorough response to my comments on the original version. I am happy with what you have done to address the issues that I originally raised and now feel that I can recommend this manuscript for publications. I think this will make an excellent contribution to this journal. Ben Woodcock

Response: Thank you for your excellent feedback; we are delighted you recognise the quality of the work and think it is suitable for publication in the journal.

Reviewer #2 (Remarks to the Author):

The response to the reviewer comments has been thorough and well thought through. Lots of changes have been made, clearly in response to diligent consideration of the reviewer comments. There are a couple of more substantive points that have not been addressed satisfactorily, and we have elaborated on them in this review. For any points from the original review we still wanted to consider at this stage, we have given the original reviewer comment, followed by author response and then our second round

40comment below. In addition we have a generic comment on the pollen vs nectar question, and two minor comments at the end.

Response: Thank you for your thorough review and explanation of your concerns. You have greatly enhanced our understanding of how exposure (and resulting risk) can be viewed, which we better articulate in the revised manuscript. We still include the nectar and pollen comparison but now justify our reason for doing so – it is the first study with spatiotemporally matched pollen and nectar samples, allowing us to confidently demonstrate that additive toxicity-scaled concentrations of residues in pollen are predictive of those in nectar. This information is helpful for pesticide residue monitoring because pollen is more humane and easier to obtain than nectar, resulting in larger sample sizes and confidence. Indeed, the European Union's Insignia project is based around a citizen-science pesticide monitoring scheme using honeybee colonies and including pollen trapping. Based on your suggested wording, we now clearly include our caveat that we do not include consumption throughout the manuscript.

Original reviewer comment:

145: What about substances that don't have LD50's. Glyphosate and most herbicides don't have LD50s, they are written as LD50>200µg a.i. per bee. How were these handled?

Response:

We now provide this information- "LD50s were rounded down to their value when expressed as 'greater than'35." – L177-178.

Second round reviewer comment:

The value of 200µg isn't an estimate, it's a wholly arbitrary value. We just stop testing at 200µg because it becomes hard to work with such concentrated substances. You need to state that this is a directional assumption i.e., the model will systematically overestimate the risk from these substances. The rounding down is fine, but given it essentially is just assigning an arbitrary value to the substance, this needs a mention that it'll overestimate risk.

Response: Thank you for bringing this to our attention. We now include the following sentence – “Finally, we used the tested dose for LD₅₀s based on limit tests³⁹ (used when a compound is expected to be low in toxicity or there are issues with solubility⁴³), which can overestimate a compound's toxicity. Three of these compounds ranked highly for compound-specific risk due to their high concentrations and frequency of detection rather than toxicity (Table 1).”– L177-181.

Original reviewer comment:

153: Given only nectar and pollen are quantified, and both represent oral exposure, it's hard to see why the contact and oral values were averaged, rather than just using the oral value. Personally, we really can't see any justification for this, as the two exposure routes are so distinct, and only oral really related to the residue work conducted. Please substantively explain to the reader why you chose to do this or use oral only. Also, would not a LC50 be more apt, the residues being returned are measured as concentrations, yet you're using acute values. Toxicologically it's a big assumption to equate acute and oral. LC50s are also widely available for Apis, again please justify or change.

We average oral and contact LD50s to provide an overall indicator of the mixture's toxicity and have clarified our rationale - " We averaged each compound's acute oral and contact LD5037 to provide an overall indicator of toxicity, reflective of how bees encounter pesticides in the landscape, i.e. moving contaminated food in contact with their bodies for oral consumption⁴⁰." – L170-172.

Unfortunately, we could not find extensive Apis LC50s and thus used LD50s from the Pesticide Properties Data Base, consistent with other researchers, e.g. Sanchez-Bayo and Goka (2014).

Second round reviewer comment:

We're not convinced by this as a justification. Bees contact exposure doesn't really happen when they move nectar/pollen back to the nest. It mostly happens when foraging and they are passing through a spray cloud or contact spray residues on surfaces. It also happens in a much much more diffuse manner when touching stuff like soil or pollen, but this is not ever going to cause acute toxicity analogous to a contact LD50. Simply, pollen isn't a source of contact exposure that translates to contact LD50s. Perhaps this wording "We averaged each compound's acute oral and contact LD5037 to provide an overall indicator of toxicity, reflective of how bees encounter pesticides in the landscape and the multiple exposure routes they can be exposed through" would more accurately surmise this. Removing the pollen example will actually strengthen the argument.

Response: Thank you, and we agree. We now use your suggestion.

Original reviewer comment:

General: The naming of the supplementary files is confusing, and I struggled to jump back and forth between them. Please consider how best to lay them out and title them from a readers perspective. The supplementary files are relied upon very heavily in a paper like this, so this is crucial.

Response:

We agree this was confusing and have worked to restructure the supplementary more logically, including a contents page with hyperlinks that meets the journal's guidelines.

Second round reviewer comment:

Maybe it's still the journal's system, but all the files came through with numerical names and we had to

open them one by one and rename them something sensible. Just have another look at the actual file names and if it's not the journal's fault give them more logical names. Maybe it is an error on our side, and apologies if so.

Response: Unfortunately, the file names result from the journal's system.

Pollen vs nectar.

Generic Second round reviewer comment addressing the Pollen vs Nectar comparison:

This is the main area we are not happy with the edits. It doesn't seem like the text has changed to reflect the points we made. Perhaps we were not clear on what we were explaining. It's only 2 sentences in the manuscript, so it's not a pervasive issue. We would advocate for just removing the two sentences entirely to avoid this issue stalling the (otherwise fabulous) paper's publication. Lines 282-285 and 340 - 342 are the sentences of interest here.

Response: Thank you for your thorough comments about comparing nectar and pollen. Below, we address your key points with specific responses.

We asked for a frank discussion of how nectar and pollen contribute to bee toxicity differently and how the modelling cannot answer this question. What has been changed amounts to some minor edits which unfortunately use dense scientific language to qualify the statements.

Put simply, we are not convinced that the pollen vs nectar claim is true, and think it is outside the scope of the model to answer this question. The attempts to qualify the claims have not been satisfactory, nor has the longer explanation in the reviewer response been sufficient to allay our concerns.

The paper claims "Environmental pesticide risk was higher in pollen than in nectar" and "we found that exposure and risk were higher in pollen than in nectar". These two claims, in our eyes, are false (or at least heavily misleading to a reader).

Firstly, the edit to change 'risk' to 'Environmental pesticide risk' is not effective in detailing what the intended meaning. The use of technical language obfuscates the point and puts the emphasis on the reader remembering the terminology and distinction, rather than spelling it out for them. If a reader does not know what the distinction between 'Environmental pesticide risk' and 'risk' is, they will assume they are the same, making the sentence untrue. Readers are seeing the paper for the first time, so it is important to assume they know nothing about it, may not even read whole sections, and thus act accordingly.

Second. To state that exposure to pesticides is higher in pollen than nectar is not true. What was found

was higher concentrations in pollen. Then the assumption of equal consumption was made. With this assumption, your statements are true, but the assumption is false. The statement "we found that exposure [was] higher in pollen" is untrue. It is possible to contest this by arguing bees face equivalent exposure to pollen and nectar. If sufficient references and data can be presented to convince us this is the case, this will suffice. But given our understanding of exposure to pesticides, we find this unlikely to happen.

Response: We now clarify that our indicator of risk is additive toxicity scaled concentrations in each section of the manuscript: Abstract L33, Introduction L46-47, Methods L167, and Discussion L290. Furthermore, we now include the caveat, that risk does not account for residue consumption when describing pollen and nectar in the Results L282-283 and Discussion L298-300, L341-344, L372-376.

The model used is fantastic at answering the central questions of the paper. However, it's not suitable for a comparison of pollen and nectar. It would need additional terms and to include the differential consumption (or an inclusive exposure term that accounts for non-consumption exposure i.e., contact toxicity).

As noted in the prior review, we do not believe it reasonable to ask for the model to be changed. We do ask that the inferences from the model be changed to fit only what is reasonable to infer from it. The pollen vs nectar' environmental pesticide risk' comparison is not reasonable to infer.

In an already very data rich and expansive paper, we see no real reason to include these datapoints. They are a methodologically flawed comparisons. Simply cut them and spend those words elaborating on the strong and rigorous conclusions. It would be possible to elaborate on and properly contextualise the results, but doing that (as we have below) raises the question of why they are interesting to include. Modifications to the text as below may help clarify this issue:

"Our model found environmental pesticide risk was higher in pollen than in nectar (Fig. 5a; $T = -10.55$, $df = 93.86$, $P < 0.01$), although this assumes equal exposure and consumption of both matrices. This assumption is unlikely to be true given bees consume very different quantities of pollen and nectar, and their ecotoxicology is very different."

"...we found that exposure and risk were higher in pollen than in nectar, although this may be a factor of the model not accounting for differential consumption/exposure to the two mediums, which are not equivalent."

Author Response:

We include the pollen and nectar comparison because our terminology reflects bees' environmental pesticide-related risk, i.e. risk outside of the nest and thus does not consider consumption, now clarified

throughout the manuscript. Quantifying the identity, level and frequency of pesticide compounds that bees encounter in this way is a step toward policy goals to reduce pesticide risk and ambitions for a more realistic risk assessment.

Second round reviewer comment:

This does not make much sense to us. How is pollen causing risk outside of the nest? It's not ingested (outside the nest) in pollen basket species, and it will cause near zero contact toxicity (it cannot penetrate the cuticle). If the paper is discussing risk outside the nest, pollen will contribute essentially no risk whatsoever. This interpretation of what the model is measuring is odd and not conveyed well in the paper.

How can risk not consider consumption? Risk = hazard (accurately quantified in your model) x Exposure. Exposure = concentration x (consumption for oral exposure OR contact amount for contact exposure). You neither quantify consumption nor contact amount. Hence risk cannot be quantified. What is then being calculated is simply hazard * concentration, which is a parameter which will align with risk, but will not allow for a medium-to-medium comparison (as medium-to-medium the consumption/contact amount changes). To call it risk is an acceptable misnomer so long as long as there is no medium-to-medium comparison which highlights the weakness of the comparison.

It feels as if there is a misunderstanding of what 'risk' means in an ecotoxicology sense, or that the redefinition of it in the paper is causing this confusion.

Response: We compare risk (toxicity-weighted concentrations) between pollen and nectar because it is helpful to know if residues in pollen are predictive of those in nectar for pesticide residue monitoring, as pollen is more straightforward to sample. However, based on your suggested wording, we now include the caveat that our risk metric does not account for consumption. Including the pollen and nectar data in this manuscript also means that if readers wish, they can estimate exposure via consumption.

To our mind, exposure - the degree to which an organism encounters pesticides at a given time and place (Sponsler et al. 2019), does not need to account for consumption. However, consumption is needed to calculate a dose (Gradish et al., 2019). Our distinction between exposure and dose also follows that of the US EPA.

Original reviewer comment:

288: Throughout there's the assumption of agriculture intensifying meaning pesticide use is going up. But pesticide use globally plateaued in 2018 (<https://www.fao.org/3/cb1329en/CB1329EN.pdf>), and in some nations like the UK it is in decline. Complicating matters is the various units of pesticide use. In Sweden it may be going up (I don't know), but the paper isn't framed around Sweden as it's framed very broadly. Readers will just accept your statement of increasing pesticide use, but this isn't conveying the

45nuanced reality, e.g. the EU has committed to a 50% reduction in pesticide use in the next decade or so.
Response:

We only discuss intensive agriculture as a practice that includes, among other things, high reliance on pesticides and reduced semi-natural habitats (Foley et al. 2005) and have checked and changed our wording throughout. For example, "However, limited foragers may become disproportionately more exposed in intensive agricultural landscapes, where there is an increased likelihood of contamination in diminished semi-natural habitats (Fig. 1b; line slope)." – L66-68.

Second round reviewer comment:

The argument that agriculture is intensifying is still an argument made in your piece. Wordings like Line 52- "Thus, as agriculture intensifies, the remaining areas of semi-natural habitats that usually provide refuge from pesticides may increasingly become a potential source of exposure" explicitly state agriculture is intensifying.

If this wording is to be retained, you need to justify this with data. However, FAO data (<https://www.fao.org/3/cb1329en/CB1329EN.pdf>) shows global pesticide use has plateaued and land area for agricultural use has declined. So we would recommend revisiting these statements; for example reference could be made just to agricultural intensification currently rather than in the future.

Response: We agree and have changed our wording, especially tense, to reflect that we are discussing agricultural intensification spatially rather than temporally. For example, "Thus, semi-natural habitats that could provide refuge from pesticides are more likely to be potential sources of exposure in intensively managed agricultural landscapes" – L53-55. And discussion – "In intensively managed agricultural landscapes with scarce semi-natural habitats and high pesticide use, *O. bicornis* is increasingly likely to forage in less-preferred mass-flowering crops and semi-natural habitats adjacent to arable land²⁸ and thus increase their pesticide exposure and risk." - L319-322.

Minor comments:

L44 – a word is missing

Response: Unfortunately, we cannot see where a word is missing.

Figure 2d: what is displayed on the X axis is not clear – consider adding in axis explanation

Response: We now clarify that the x-axis is 'site'. – "Non-agricultural species often dominated pollen use at each site (x-axis) (d)" – L644-645.

Decision Letter, second revision:

14th November 2022

Dear Dr. Knapp,

Thank you for submitting your revised manuscript "Ecological traits interact with landscape context to determine bees' pesticide risk" (NATECOLEVOL-220416385B). It has now been seen again by the original reviewers and their comments are below. The reviewers find that the paper has improved in revision, and therefore we'll be happy in principle to publish it in Nature Ecology & Evolution, pending minor revisions to satisfy the reviewers' final requests and to comply with our editorial and formatting guidelines.

[REDACTED]

Reviewer #2 (Remarks to the Author):

The comments have been addressed thoroughly and throughout, and I look forward to seeing this paper published.

Our ref: NATECOLEVOL-220416385B

30th November 2022

Dear Dr. Knapp,

Thank you for your patience as we've prepared the guidelines for final submission of your Nature Ecology & Evolution manuscript, "Ecological traits interact with landscape context to determine bees' pesticide risk" (NATECOLEVOL-220416385B). Please carefully follow the step-by-step instructions provided in the attached file, and add a response in each row of the table to indicate the changes that you have made. Please also check and comment on any additional marked-up edits we have proposed within the text. Ensuring that each point is addressed will help to ensure that your revised manuscript can be swiftly handed over to our production team.

****We would like to start working on your revised paper, with all of the requested files and forms, as soon as possible (preferably within two weeks). Please get in contact with us immediately if you anticipate it taking more than two weeks to submit these revised files.****

In recognition of the time and expertise our reviewers provide to Nature Ecology & Evolution's editorial process, we would like to formally acknowledge their contribution to the external peer review of your manuscript entitled "Ecological traits interact with landscape context to determine bees' pesticide risk". For those reviewers who give their assent, we will be publishing their names alongside the published article.

Nature Ecology & Evolution offers a Transparent Peer Review option for new original research manuscripts submitted after December 1st, 2019. As part of this initiative, we encourage our authors to support increased transparency into the peer review process by agreeing to have the reviewer comments, author rebuttal letters, and editorial decision letters published as a Supplementary item. When you submit your final files please clearly state in your cover letter whether or not you would like to participate in this initiative. Please note that failure to state your preference will result in delays in accepting your manuscript for publication.

48Cover suggestions

As you prepare your final files we encourage you to consider whether you have any images or illustrations that may be appropriate for use on the cover of Nature Ecology & Evolution.

Nature Ecology & Evolution has now transitioned to a unified Rights Collection system which will allow our Author Services team to quickly and easily collect the rights and permissions required to publish your work. Approximately 10 days after your paper is formally accepted, you will receive an email in providing you with a link to complete the grant of rights. If your paper is eligible for Open Access, our Author Services team will also be in touch regarding any additional information that may be required to arrange payment for your article.

Please note that *Nature Ecology & Evolution* is a Transformative Journal (TJ). Authors may publish their research with us through the traditional subscription access route or make their paper immediately open access through payment of an article-processing charge (APC). Authors will not be required to make a final decision about access to their article until it has been accepted. [Find out more about Transformative Journals](https://www.springernature.com/gp/open-research/transformative-journals)

Authors may need to take specific actions to achieve [compliance with funder and institutional open access mandates](https://www.springernature.com/gp/open-research/funding/policy-compliance-faqs). If your research is supported by a funder that requires immediate open access (e.g. according to [Plan S principles](https://www.springernature.com/gp/open-research/plan-s-compliance)) then you should select the gold OA route, and we will direct you to the compliant route where possible. For authors selecting the subscription publication route, the journal's standard licensing terms will need to be accepted, including [self-archiving-and-license-to-publish](https://www.nature.com/nature-portfolio/editorial-policies/self-archiving-and-license-to-publish). Those licensing terms will supersede any other terms that the author or any third party may assert apply to any version of the manuscript.nature portfolio

For information regarding our different publishing models please see our [Transformativ Journals](https://www.springernature.com/gp/open-research/transformative-journals) page. If you have any questions about costs, Open Access requirements, or our legal forms, please contact ASJournals@springernature.com.

[REDACTED]

[REDACTED]

Reviewer #4:

Remarks to the Author:

The comments have been addressed thoroughly and throughout, and I look forward to seeing this paper published.

Final Decision Letter:

22nd December 2022

Dear Dr Knapp,

We are pleased to inform you that your Article entitled "Ecological traits interact with landscape context to determine bees' pesticide risk", has now been accepted for publication in Nature Ecology & Evolution.

Over the next few weeks, your paper will be copyedited to ensure that it conforms to Nature Ecology and Evolution style. Once your paper is typeset, you will receive an email with a link to choose the appropriate publishing options for your paper and our Author Services team will be in touch regarding any additional information that may be required

You will not receive your proofs until the publishing agreement has been received through our system

50Due to the importance of these deadlines, we ask you please us know now whether you will be difficult to contact over the next month. If this is the case, we ask you provide us with the contact information (email, phone and fax) of someone who will be able to check the proofs on your behalf, and who will be available to address any last-minute problems . Once your paper has been scheduled for online publication, the Nature press office will be in touch to confirm the details.

Acceptance of your manuscript is conditional on all authors' agreement with our publication policies (see www.nature.com/authors/policies/index.html). In particular your manuscript must not be published elsewhere and there must be no announcement of the work to any media outlet until the publication date (the day on which it is uploaded onto our web site).

Please note that *Nature Ecology & Evolution* is a Transformative Journal (TJ). Authors may publish their research with us through the traditional subscription access route or make their paper immediately open access through payment of an article-processing charge (APC). Authors will not be required to make a final decision about access to their article until it has been accepted. [Find out more about Transformative Journals](https://www.springernature.com/gp/open-research/transformative-journals)

Authors may need to take specific actions to achieve [compliance with funder and institutional open access mandates](https://www.springernature.com/gp/open-research/funding/policy-compliance-faqs). If your research is supported by a funder that requires immediate open access (e.g. according to [Plan S principles](https://www.springernature.com/gp/open-research/plan-s-compliance)) then you should select the gold OA route, and we will direct you to the compliant route where possible. For authors selecting the subscription publication route, the journal's standard licensing terms will need to be accepted, including [self-archiving-and-license-to-publish](https://www.nature.com/nature-portfolio/editorial-policies/self-archiving-and-license-to-publish). Those licensing terms will supersede any other terms that the author or any third party may assert to any version of the manuscript.

nature portfolio

We welcome the submission of potential cover material (including a short caption of around 40 words) related to your manuscript; suggestions should be sent to Nature Ecology & Evolution as electronic files (the image should be 300 dpi at 210 x 297 mm in either TIFF or JPEG format). Please note that such pictures should be selected more for their aesthetic appeal than for their scientific content, and that colour images work better than black and white or grayscale images. Please do not try to design a cover with the Nature Ecology & Evolution logo etc., and please do not submit composites of images related to your work. I am sure you will understand that we cannot make any promise as to whether any of your suggestions might be selected for the cover of the journal.

You can generate the link yourself when you receive your article DOI by entering it here: <http://authors.springernature.com/share>.

[REDACTED]

P.S. Click on the following link if you would like to recommend Nature Ecology & Evolution to your librarian <http://www.nature.com/subscriptions/recommend.html#forms>

** Visit the Springer Nature Editorial and Publishing website at http://editorial-jobs.springernature.com?utm_source=ejp_NEcoE_email&utm_medium=ejp_NEcoE_email&utm_campaign=ejp_NEcoE for more information about our career opportunities. If you have any questions please click [here](mailto:editorial.publishing.jobs@springernature.com).**